# Invisible Image Watermarks Are Provably Removable Using Generative AI

**Xuandong Zhao**[*]
UC Berkeley

**Kexun Zhang**[*]
Carnegie Mellon University

**Zihao Su**
UC Santa Barbara

**Saastha Vasan**
UC Santa Barbara

**Ilya Grishchenko**
UC Santa Barbara

**Christopher Kruegel**
UC Santa Barbara

**Giovanni Vigna**
UC Santa Barbara

**Yu-Xiang Wang**
UC San Diego

**Lei Li**
Carnegie Mellon University

## Abstract

Invisible watermarks safeguard images' copyrights by embedding hidden messages only detectable by owners. They also prevent people from misusing images, especially those generated by AI models. We propose a family of regeneration attacks to remove these invisible watermarks. The proposed attack method first adds random noise to an image to destroy the watermark and then reconstructs the image. This approach is flexible and can be instantiated with many existing image-denoising algorithms and pre-trained generative models such as diffusion models. Through formal proofs and extensive empirical evaluations, we demonstrate that pixel-level invisible watermarks are vulnerable to this regeneration attack. Our results reveal that, across four different pixel-level watermarking schemes, the proposed method consistently achieves superior performance compared to existing attack techniques, with lower detection rates and higher image quality. However, watermarks that keep the image semantically similar can be an alternative defense against our attacks. Our finding underscores the need for a shift in research/industry emphasis from invisible watermarks to semantic-preserving watermarks. Code is available at https://github.com/XuandongZhao/WatermarkAttacker.

## 1 Introduction

Generative models like DALL-E [45], Imagen [50], and Stable Diffusion [47] can produce images that are often visually indistinguishable from those created by human photographers and artists, potentially leading to misunderstandings and false beliefs due to their visual similarity. To address this, government leaders [30, 26, 29] advocate for the responsible use of AI, emphasizing the importance of identifying AI-generated content. In response, leading AI companies such as Google [20], Microsoft [61], and OpenAI [4] have pledged to incorporate *watermarks* into their AI-generated images. *Invisible* watermarks [62, 17, 9] are preferred as they preserve image quality and are less likely to be removed by laypersons. However, abusers are aware of these watermarks and may attempt to remove them, making it crucial for invisible watermarks to be robust against evasion attacks.

Existing attacks can be categorized into two types. The first is *destructive*, where the watermark is removed by corrupting the image. Typical destructive attacks include modifying the brightness, JPEG compression, and adding Gaussian noise. These approaches are effective at removing watermarks, but

---

[*]Equal contribution. Email addresses: xuandongzhao@berkeley.edu kexunz@andrew.cmu.edu

38th Conference on Neural Information Processing Systems (NeurIPS 2024).

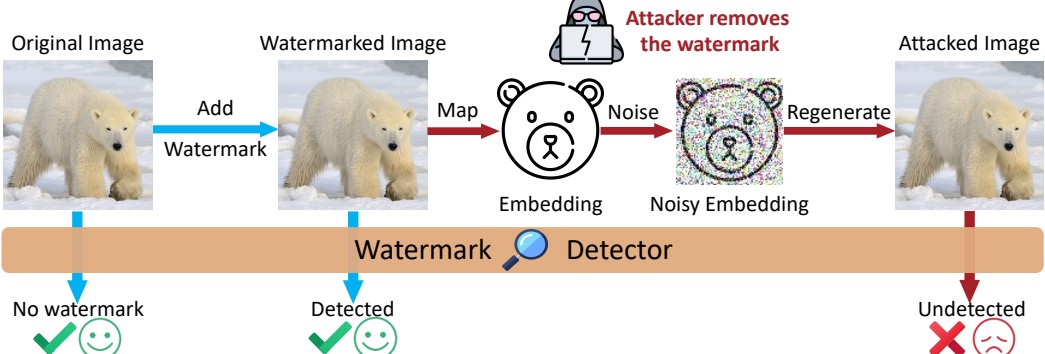

Figure 1: Removing invisible watermarks: The proposed attack first maps the watermarked image to its embedding, which is another representation of the image. Then the embedding is noised to destruct the watermark. After that, a regeneration algorithm reconstructs the image from the noisy embedding.

they result in significant quality loss. The second type of attack is *constructive*, where the watermark is treated as some noise on the original image and removed by purifying the image. Constructive attacks include image-denoising techniques like Gaussian blur [25], BM3D [11], and learning-based approaches [65]. However, they cannot remove resilient watermarks easily. To counter these attacks, learning-based watermarking methods [67, 66, 15] were proposed to explicitly train against the known attacks to be robust. But how about other attacks? What is the end of this cat-and-mouse game? In this paper, we ask a more fundamental question:

*Is an invisible watermark* necessarily *non-robust?*

To be more precise, is there a fundamental trade-off between the invisibility of a watermark and its resilience to any attack that preserves the image quality to a certain-level?

To address this question, we propose a regeneration attack that leverages the strengths of both destructive and constructive approaches. The pipeline of the attack is given in Figure 1. Our attack first corrupts the image by adding Gaussian noise to its latent representation. Then, we reconstruct the image from the noisy embedding using a generative model. The proposed regeneration attack is flexible in that it can be instantiated with various regeneration algorithms, including traditional denoisers and deep generative models such as diffusion models [23]. Ironically, the recent advances in generative models that created the desperate need for invisible watermarks are also making watermark removal easier when integrated into the proposed regeneration attack.

Surprisingly, we prove that the proposed attack guarantees the removal of certain invisible watermarks such that *no* detection algorithm could work. In other words, our attack is provably effective in removing any watermarks that perturb the image within a limited range of $\ell_2$-distance, regardless of whether they have been proposed or have not yet been invented. We also show that our attack maintains image quality comparable to the unwatermarked original.

To validate our theory, we conduct extensive experiments on four widely used invisible watermarks [41, 67, 53, 15]. Our proposed attack method significantly outperforms five baseline methods in terms of both image quality and watermark removal. For the resilient watermark scheme RivaGAN, our regeneration attacks successfully remove 98% of the invisible watermarks while maintaining a PSNR above 30 compared to the original images. With the empirical results and the theoretical guarantee, we claim that pixel-level invisible image watermarks are vulnerable to our regeneration attacks.

Given the vulnerability of invisible watermarks, we explore another option: *semantic* watermarks. Semantic watermarks do not limit the perturbation to be within an $\ell_2$-distance. As long as the watermarked image looks similar and contains similar content, it is considered suitable for use. One instance of such semantic watermarks is Tree-Ring [60], which has shown resilience against our attacks. While not a perfect solution, as the watermark becomes somewhat "visible", semantic watermarking offers a potential path forward for protecting the proper use of AI-generated images when invisible watermarks are provably ineffective.

Our contributions can be summarized as follows:

- We propose a family of regeneration attacks for image watermark removal that can be instantiated with many existing denoising algorithms and generative models.
- We prove that the proposed attack is guaranteed to remove certain pixel-based invisible watermarks and that the regenerated images are close to the original unwatermarked image.
- We evaluate the proposed attack on various invisible watermarks to demonstrate their vulnerability and its effectiveness compared with strong baselines.
- We explore other possibilities to embed watermarks in a visible yet semantically similar way. Empirical results indicate that this approach works better under our attack and is worth investigating as an alternative.

## 2 Related Work and Background

### 2.1 Related Work

**Image watermarking and steganography.** Steganography and invisible watermarking are key techniques in information hiding, serving diverse purposes such as copyright protection, privacy-preserved communication, and content provenance. Early works in this area employ hand-crafted methods, such as Least Significant Bit (LSB) embedding [62], which subtly hides data in the lowest order bits of each pixel in an image. Over time, numerous techniques have been developed to imperceptibly embed secrets in the spatial [17] and frequency [24, 44] domains of an image. Additionally, the emergence of deep learning has contributed significantly to this field. Deep learning methods offer improved robustness against noise while maintaining the quality of the generated image. SteganoGAN [66] uses generative adversarial networks (GAN) for steganography and perceptual image optimization. RivaGAN [67], further improves GAN-based watermarking by leveraging attention mechanisms. SSL watermarking [15], trained with self-supervision, enhances watermark features through data augmentation. Stable Signature [14] fine-tunes the decoder of Latent Diffusion Models to add the watermark. Tree-Ring [60] proposes a semantic watermark, which watermarks generative diffusion models using minimal shifts of their output distribution. This work focuses on the removal of invisible watermarks, as opposed to visible watermarks, for several reasons. Visible watermarks are straightforward for potential adversaries to visually identify and locate within an image. Additionally, removal techniques for visible watermarks are already extensively studied in prior work such as [35, 21, 10]. In contrast, invisible watermarks do not have explicit visual cues that reveal their presence or location. Developing removal techniques for imperceptible embedded watermarks presents unique challenges.

**Deep generative models.** The high-dimensional nature of images poses unique challenges to generative modeling. In response to these challenges, several types of deep generative models have been developed, including Variational Auto-Encoders (VAEs) [56, 55], Generative Adversarial Networks (GANs) [18], flow-based generative models [46], and diffusion models [23, 47]. These models leverage deep latent representations to generate high-quality synthetic images and approximate the true data distribution. One particularly interesting use of generative models is for data purification, i.e., removing the adversarial noise from a data sample. The purification is similar to watermark removal except that purification is a defense strategy while watermark removal is an attack. The diffusion-based approach in [43] is similar to an instance of our regeneration attack, but the usage is different in our paper and our theoretical guarantee of watermark removal is stronger. In this paper, we aim to demonstrate the capability of these deep generative models in removing invisible watermarks from images by utilizing the latent representations obtained through the encoding and decoding processes.

### 2.2 Problem Setup

This section defines *invisible* watermarks and the properties of an algorithm for their detection. It then discusses the threat model for the removal of invisible watermarks.

**Definition 2.1** (Invisible watermark). Let $x \in \mathcal{X}$ be the original image and $x_w = \mathsf{Watermark}(x, \mathsf{aux})$ be the watermarked image for a watermarking scheme that is a function of $x$ and any auxiliary information aux, e.g., a secret key. We say that the watermark is $\Delta$-*invisible* on a clean image $x$ w.r.t. a "distance" function $\mathsf{dist} : \mathcal{X} \times \mathcal{X} \to \mathbb{R}_+$ if $\mathsf{dist}(x, x_w) \leq \Delta$.

**Definition 2.2** (Watermark detection). A watermark detection algorithm Detect : $\mathcal{X} \times \text{aux} \to \{0,1\}$ determines whether an image $\tilde{x} \in \mathcal{X}$ is watermarked with the auxiliary information (secret key) being aux. Detect may make two types of mistakes, false positives (classifying an unwatermarked image as watermarked) and false negatives (classifying a watermarked image as unwatermarked). $\tilde{x}$ could be drawn from either the null distribution $P_0$ or watermarked distribution $P_1$. We define Type I error (or false positive rate) $\epsilon_1 := \Pr_{x \sim P_0}[\text{Detect}(x) = 1]$ and Type II error (or false negative rate) $\epsilon_2 := \Pr_{x \sim P_1}[\text{Detect}(x) = 0]$.

A watermarking scheme is typically designed such that $P_1$ is different from $P_0$ so the corresponding (carefully designed) detection algorithm can distinguish them almost perfectly, that is, to ensure $\epsilon_1$ and $\epsilon_2$ are nearly 0. An attack on a watermarking scheme aims at post-processing a possibly watermarked image which changes both $P_0$ and $P_1$ with the hope of increasing Type I and Type II error at the same time, hence evading the detection.

We consider the following threat model for removing invisible watermarks from images:

**Adversary's capabilities.** We assume that an adversary only has access to the watermarked images. The watermarking scheme Watermark, the auxiliary information aux, and the detection algorithm Detect are unknown to the adversary. The adversary can make modifications to these already watermarked images it has access to using arbitrary side information and computational resources, but it cannot rely on any specific property of the watermarking process and it cannot query Detect.

**Adversary's objective.** The primary objective of an adversary is to render the watermark detection algorithm ineffective. Specifically, the adversary aims to produce an image $\tilde{x}$ from the watermarked image $x_w$ which causes the Detect algorithm to always have a high Type I error (false positive rate) or a high Type II error (false negative rate). Simultaneously, the output image $\tilde{x}$ should maintain comparable quality to the original, non-watermarked image. The adversary's objective will be formally defined later.

### 2.3 Invisible Watermark Detection

Watermarking methods implant $k$-bit secret information into images. The detection algorithm Detect uses an extractor for this hidden data, applying a statistical test to ascertain if the extracted message matches the secret. This test evaluates the number of matching bits $M(m, m')$ between the extracted $m'$ and original $m \in \{0,1\}^k$ messages. A watermark is detected if $M(m, m') \geq \tau$ for a predefined threshold $\tau$ [63, 38]. Formally, we distinguish whether an image $x$ was watermarked ($H_1$) or not ($H_0$). Assuming under null hypothesis $H_0$ that the extracted bits behave like independent Bernoulli variables with a 0.5 probability, $M(m, m')$ follows a binomial distribution $B(k, 0.5)$. The false positive rate ($\epsilon_1$) corresponds to the probability that $M(m, m')$ exceeds $\tau$. This has a closed form using the regularized incomplete beta function $I_x(a; b)$:

$$\epsilon_1(\tau) = \mathbb{P}\left(M\left(m, m'\right) > \tau \mid H_0\right) = \frac{1}{2^k} \sum_{i=\tau+1}^{k} \binom{k}{i} = I_{1/2}(\tau + 1, k - \tau).$$

**Decision threshold.** We consider a watermark to be detected, if we can reject the null hypothesis $H_0$ with a $p$-value less than 0.01. In practice, for a $k = 32$-bit watermark, we require at least 23 bits to be extracted correctly in order to confirm the presence of a watermark. This provides a reasonable balance between detecting real watermarks and avoiding false positives.

## 3 The Proposed Regeneration Attack

Our attack method first destructs a watermarked image by adding noise to its representation, and then reconstructs it from the noised representation.

Specifically, given an embedding function $\phi : \mathbb{R}^n \to \mathbb{R}^d$, a regeneration function $\mathcal{A} : \mathbb{R}^d \to \mathbb{R}^n$, and a noise level $\sigma$, the attack algorithm takes a watermarked image $x_w \in \mathbb{R}^n$ and returns

$$\hat{x} = \underbrace{\mathcal{A}\Big( \overbrace{\phi(x_w) + \mathcal{N}(0, \sigma^2 I_d)}^{\text{destructive}} \Big)}_{\text{constructive}}. \tag{1}$$

**Algorithm 1** Regeneration Attack Instance: Removing invisible watermarks with a diffusion model

---

**input** The watermarked image $x_w$, a time step $t^*$ determining the level of noise added.
**output** A reconstructed clean image $\hat{x}$.
 1: $z_0 \leftarrow \phi(x_w)$ // map the watermarked image $x_w$ to latent space
 2: $\epsilon \sim \mathcal{N}(0, I_d)$ // sample a random normal Gaussian noise
 3: $z_{t^*} \leftarrow \sqrt{\alpha(t^*)}z_0 + \sqrt{1-\alpha(t^*)}\epsilon$ // add noise to the latent, noise level determined by $t^*$
 4: $\hat{z}_0 \leftarrow \mathsf{solve}(z_{t^*}, t^*, s, f, g)$ // denoise the noised latent to reconstruct a clean latent
 5: $\hat{x} \leftarrow \theta(\hat{z}_0)$ // map the reconstructed latent back to a watermark-free image
 6: **return** $\hat{x}$

---

The first step of the algorithm is *destructive*. It maps the watermarked image $x_w$ to an embedding $\phi(x_w)$ (which is a possibly different representation of the image), and adds i.i.d. Gaussian noise. The explicit noise shows the destructive nature of the first step. The second step of the algorithm is *constructive*. The corrupted image representation $\phi(x_w) + \mathcal{N}(0, \sigma^2 I_d)$ is passed through a regeneration function $\mathcal{A}$ to reconstruct the original clean image.

There are various different choices for $\phi$ and $\mathcal{A}$ that can instantiate our attack. $\phi$ can be as simple as identity map, or as complicated as deep generative models including variational autoencoders [31]. $\mathcal{A}$ can be traditional denoising algorithms from image processing and recent AI models such as diffusion [23]. The choice of $\phi$ and $\mathcal{A}$ may change the empirical results, but it does not affect the theoretical guarantee. In the following sections, we introduce three combinations of $\phi$ and $\mathcal{A}$ to instantiate the attack. Among the three, the diffusion instantiation is the most complicated and we describe it with pseudocode in Algorithm 1.

### 3.1 Attack Instance 1: Identity Embedding with Denoising Reconstruction

Set $\phi$ to be identity map, then $\mathcal{A}$ can be any image denoising algorithm, e.g., BM3D [11], TV-denoising [48], bilateral filtering [54], DnCNNs [65], or a learned natural image manifold [6, 11, 65, 64]. A particular example of interest is a "denoising autoencoder" [57], which takes $\phi$ to be identity, adds noise to the image deliberately, and then denoises by attempting to reconstruct the image. Observe that for "denoising autoencoder" we do not need to add additional noise.

### 3.2 Attack Instance 2: VAE Embedding and Reconstruction

The regeneration attack in Equation 1 can be instantiated with a variational autoencoder (VAE). A VAE [31] consists of an encoder $q_\phi(z|x)$ that maps a sample $x$ to the latent space $z$ and a decoder $p_\theta(x|z)$ that maps a latent $z$ back to the data space $x$. Both the encoder and decoder are parameterized with neural networks. VAEs are trained with a reconstruction loss that measures the distance from the reconstructed sample to the original sample and a prior matching loss that restricts the latent to follow a pre-defined prior distribution.

Instead of mapping $x$ directly to $z$, the encoder maps it to the mean $\mu(x)$ and variance $\sigma(x)$ of a Gaussian distribution and samples from it. Therefore, VAE already adds noise during the encoding stage (though its variance depends on the sample $x$, which is not exactly the same as defined in Equation 1), so there is no need to add extra noise. Note that this is similar to the situation of denoising autoencoders described in Section 3.1, as the denoising autoencoder is a trivial case of VAE where $\mu(x)$ is identity.

### 3.3 Attack Instance 3: Diffusion Embedding and Reconstruction

The regeneration attack can also be instantiated with diffusion models. Diffusion models [23] define a generative process that learns to sample from an unknown true distribution $p(z_0)$. This process is learned by estimating original samples from randomly noised ones. In other words, diffusion models are trained to denoise, which makes them candidates for the regeneration function $\mathcal{A}$ in the proposed attack. The embedding function $\phi$ can either be identity [23] or a latent embedding [47], depending on the space where diffusion is trained.

For diffusion models, the process of adding noise to a clean sample is known as the *forward process*. Likewise, the process of denoising a noisy sample is known as the *backward process*.

The forward process is defined by the following stochastic differential equation (SDE): $dz = f(z, t)dt + g(t)dw$ (2), where $t \in [0, 1], z \in \mathbb{R}^d$, $w(t) \in \mathbb{R}^d$ is a standard Wiener process, and $f, g$ are real-valued functions. The backward process can then be described with its reverse SDE: $d\hat{z} = \left[ f(\hat{z}, t) - g(t)^2 \nabla_{\hat{z}} \log p_t(\hat{z}) \right] dt + g(t)d\bar{w}$ (3), where $\hat{w}$ is a reverse Wiener process. Diffusion models parameterize $\nabla_{\hat{z}} \log p_t(\hat{z})$ with a neural network $s(z, t)$. By substituting $s(z, t)$ into Equation 3, the backward SDE becomes known and solvable using numerical solvers [34, 36], $\hat{z}_0 = \text{solve}(z_t, t, s, f, g)$.

Among many ways to define $f$ and $g$ in Equation 2, variance preserving SDE (VP-SDE) is commonly used [23, 47]. Under this setting, the conditional distribution of the noised sample is the following Gaussian [51]: $p(z_t|z_0) = \mathcal{N}(\sqrt{\alpha(t)}z_0, 1 - \alpha(t))$ (4), where $\alpha(t)$ a pre-defined noise schedule. The variance of the original distribution $p(z_0)$ is preserved at any step.

As defined in Algorithm 1, our algorithm removes the watermark from the watermarked image $x_w$ using diffusion models. $x_w$ is first mapped to the latent representation $z_0$, which is then noised to the time step $t^*$. A latent diffusion model is then used to reconstruct the latent $\hat{z}_0$, which is mapped back to an image $\hat{x}$.

Similar to denoising autoencoders, in either diffusion or VAEs, the noise-injection is integral to the algorithms themselves, and no additional noise-injection is needed.

# 4 Theoretical Analysis

We show in this section that the broad family of regeneration attacks as defined in Equation 1 enjoy provable guarantees on their ability to remove invisible watermarks while retaining the high quality of the original image. Our proofs are deferred to Appendix D. More discussion on the implications and interpretation of our theoretical analysis can be found in Appendix C.

## 4.1 Certified Watermark Removal

How do we quantify the ability of an attack algorithm to remove watermarks? We argue that if after the attack, no algorithm is able to distinguish whether the result is coming from a watermarked image or the corresponding original image without the watermark, then we consider the watermark certifiably removed. More formally:

**Definition 4.1** ($f$-Certified-Watermark-Free). We say that a watermark removal attack is $f$-Certified-Watermark-Free (or $f$-CWF) against a watermark scheme for a non-increasing function $f : [0, 1] \to [0, 1]$, if for any detection algorithm Detect : $\mathcal{X} \times \text{aux} \to \{0, 1\}$, the Type II error (false negative rate) $\epsilon_2$ of Detect obeys that $\epsilon_2 \geq f(\epsilon_1)$ for all Type I error $0 \leq \epsilon_1 \leq 1$.

Let us also define a parameter to quantify the effect of the embedding function $\phi$.

**Definition 4.2** (Local Watermark-Specific Lipschitz property). We say that an embedding function $\phi : \mathcal{X} \to \mathbb{R}^d$ satisfies $L_{x,w}$-Local Watermark-Specific Lipschitz property if for a watermark scheme $w$ that generates $x_w$ with $x$,

$$\|\phi(x_w) - \phi(x)\| \leq L_{x,w}\|x_w - x\|.$$

The parameter $L_{x,w}$ measures how much the embedding compresses the watermark added on a particular clean image $x$. If $\phi$ is identity, then $L_{x,w} \equiv 1$. If $\phi$ is a projection matrix to a linear subspace then $0 \leq L_{x,w} \leq 1$ depending on the magnitude of the component of $x_w - x$ in this subspace. For a neural image embedding $\phi$, the exact value of $L_{x,w}$ is unknown but given each $x_w$ and $x$ it can be computed and estimated efficiently. We defer more discussion to the Appendix A.

**Theorem 4.3.** *For a $\Delta$-invisible watermarking scheme with respect to $\ell_2$-distance. Assume the embedding function $\phi$ of the diffusion model is $L_{x,w}$-Locally Lipschitz. The randomized algorithm $\mathcal{A}(\phi(\cdot) + \mathcal{N}(0, \sigma^2 I_d))$ produces a reconstructed image $\hat{x}$ which satisfies $f$-CWF with*

$$f(\epsilon_1) = \Phi\left(\Phi^{-1}(1 - \epsilon_1) - \frac{L_{x,w}\Delta}{\sigma}\right),$$

*where $\Phi$ is the Cumulative Density Function function of the standard normal distribution.*

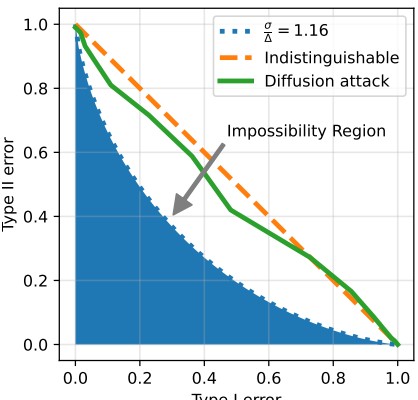

Figure 2: Theoretical and empirical trade-off functions for DwtDctSvd watermark detectors after our attack. Trade-off functions indicate how much less Type II error (false negative rate) the detector gets in return by having more Type I error (false positive rate). Theoretically, after the attack, no detection algorithm can fall in the *Impossibility Region* and have both Type I error and Type II error at a low level. Empirically, the watermark detector performs even worse than the theory, indicating the success of our attack and the validity of the theoretical bound. We use 500 watermarked MS-COCO images with an empirically valid upper bound of $L = 1$ and noise level $\sigma = 1.16\Delta$. An additional example for the RivaGAN watermark is provided in Figure 12.

Figure 2 illustrates what the tradeoff function looks like. The result says that after the regeneration attack, it is impossible for any detection algorithm to correctly detect the watermark with high confidence. In addition, it shows that such detection is as hard as telling the origin of a single sample $Y$ from either of the two Gaussian distributions $\mathcal{N}(0, 1)$ and $\mathcal{N}(L_{x,w}\Delta/\sigma, 1)$. Specifically, when there is a uniform upper bound $L \geq L_{x,w}$, we can calibrate $\sigma$ such that the attack is provably effective for a *prescribed* $\Phi(\Phi^{-1}(1 - \cdot) - L\Delta/\sigma)$-CWF guarantee for *all* input images and *all* $\Delta$-invisible watermarks (see specific constructions in Appendix A).

The proof, deferred to Appendix D, leverages an interesting connection to a modern treatment of differential privacy [13] known as the Gaussian differential privacy [12]. The work of [12] itself is a refinement and generalization of the pioneering work of [59] and [28] which established a tradeoff-function view.

## 4.2   Utility Guarantees

In this section, we prove that the regenerated image $\hat{x}$ is close to the original (unwatermarked) image $x_0$. This is challenging because the denoising algorithm only gets access to the noisy version of the watermarked image. Interestingly, we can obtain a general extension lemma showing that for any black-box generative model that can successfully denoise a noisy yet unwatermarked image with high probability, the same result also applies to the watermarked counterpart, except that the failure probability is slightly larger.

**Theorem 4.4.** *Let $x_0$ be an image with $n$ pixels and $\phi : \mathbb{R}^n \to \mathbb{R}^d$ be an embedding function. Let $\mathcal{A}$ be an image generation / denoising algorithm such that with probability at least $1 - \delta$, $\|\mathcal{A}(\phi(x_0) + \mathcal{N}(0, \sigma^2 I_d)) - x_0\| \leq \xi_{x_0,\sigma,\delta}$. Then for any $\Delta$-invisible watermarking scheme that produces $x_w$ from a clean image $x_0$, then $\hat{x} = \mathcal{A}(\phi(x_w) + \mathcal{N}(0, \sigma^2 I_d))$ satisfies that*

$$\|\hat{x} - x_0\| \leq \xi_{x_0,\sigma,\delta}$$

*with a probability at least $1 - \tilde{\delta}$, where $\tilde{\delta} = \min_{v \in \mathbb{R}} \left\{ \delta \cdot e^v + \Phi\left(\frac{\tilde{\Delta}}{2\sigma} - \frac{v\sigma}{\tilde{\Delta}}\right) - e^v \Phi\left(-\frac{\tilde{\Delta}}{2\sigma} - \frac{v\sigma}{\tilde{\Delta}}\right) \right\}$ in which $\Phi$ denotes the standard normal CDF and $\tilde{\Delta} := L_{x_0,w}\Delta$.*

The theorem says that if a generative model is able to denoise a noisy version of the original image, then the corresponding watermark-removal attack using this generative model provably produces an image with similar quality.

**Corollary 4.5.** *The expression for $\tilde{\delta}$ above can be (conservatively) simplified to $\tilde{\delta} \leq e^{\frac{L_{x,w}^2 \Delta^2}{\sigma^2}} \cdot \delta^{1/2}$.*

*For example if $\sigma \asymp L_{x,w}\Delta$, then this is saying that if $\xi_{x_0,\sigma,\delta}$ depends logarithmically on $1/\delta$, the same exponential tail holds for denoising the watermarked image.*

The above result is powerful in that it makes no assumption about what perturbation the watermarking schemes could inject and which image generation algorithm we use. We give a few examples below.

For denoising algorithms with theoretical guarantees, e.g., TV-denoising [27, Theorem 2], our results imply provable guarantees on the utility for the watermark removal attack of the form, "w.h.p., $\frac{1}{n}\|\hat{x} - x_0\|^2 = \tilde{O}\left(\frac{\sigma \text{TV}_{2d}(x_0)}{n}\right)$", i.e., vanishing mean square error (MSE) as $n$ gets bigger.

| Original Image | DwtDctSvd WM | Brightness | JPEG | Gaussian blur | **VAE attack** | **Diffusion attack** |

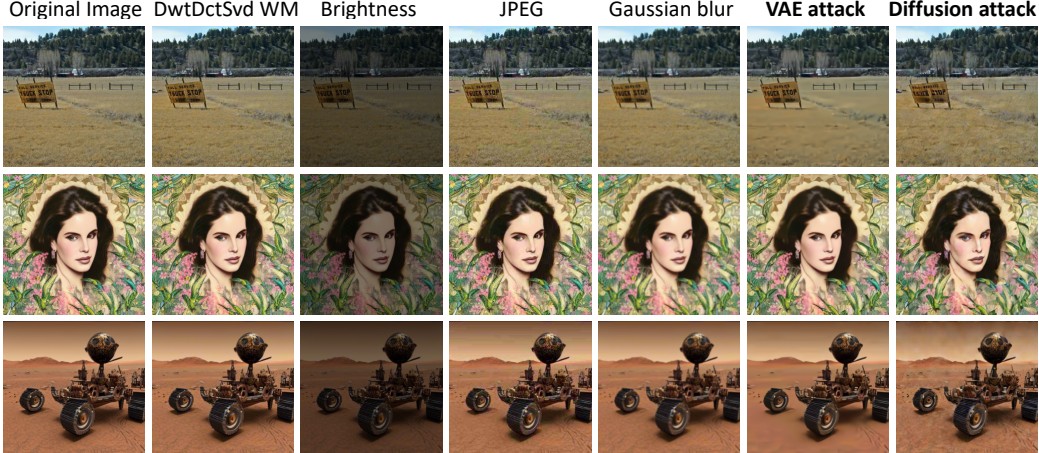

Figure 3: Examples of attacks on DwtDctSvd watermarking, including destructive attacks (e.g., brightness change and JPEG compression), constructive attacks (e.g., Gaussian blur), and regeneration attacks using VAEs and diffusion models. Brightness change, JPEG compression, VAE attack, and diffusion attack successfully remove the watermark. The VAE attack over-smooths the image, resulting in blurriness. The diffusion attack maintains high image quality while removing the watermark. Additional attack examples for other watermarking schemes are in Figures 8, 9, 10, 11.

For modern deep learning-based image denoising and generation algorithms where worst-case guarantees are usually intractable, Theorem 4.4 is still applicable for each image separately. That is to say, as long as their empirical denoising quality is good on an unwatermarked image, the quality should also be good on its watermarked counterpart.

## 5 Evaluation

**Datasets.** We evaluate our attack on two types of images: real photos and AI-generated images. For real photos, we use 500 randomly selected images from the MS-COCO dataset [33]. For AI-generated images, we employ the `Stable Diffusion-v2.1` model[2] from Stable Diffusion [47], a state-of-the-art generative model capable of producing high-fidelity images. Using prompts from the Stable Diffusion Prompt (SDP) dataset[3], we generate 500 images encompassing both photorealistic and artistic styles. This diverse selection allows for a comprehensive evaluation of our attack on invisible watermarks. All experiments are conducted on Nvidia A6000 GPUs.

**Watermark settings.** We evaluate four publicly available pixel-level watermarking methods: DwtDctSvd [41], RivaGAN [67], StegaStamp [53], and SSL watermark [15]. These methods represent a variety of approaches, ranging from traditional signal processing to recent deep learning techniques, as introduced in Section 2.1. To account for watermarks of different lengths, we use $k = 32$ bits for DwtDctSvd, RivaGAN, and SSL watermark, and $k = 96$ bits for StegaStamp. For watermark detection, we set the decision threshold to reject the null hypothesis with $p < 0.01$, requiring the detection of 23 out of 32 bits or 59 out of 96 bits, respectively, for the corresponding methods, as described in Section 2.3. For watermark extraction, we use the publicly available code for each method with default inference and fine-tuning parameters specified in their papers.

**Attack baselines.** To thoroughly evaluate the robustness of our proposed watermarking method, we test it against a diverse set of baseline attacks representing common image perturbations. We select both geometric/quality distortions and noise manipulations that could potentially interfere with embedded watermarks. Specifically, the baseline attack set includes: brightness adjustments with enhancement factors of [2, 4, 6, 8, 12], contrast adjustments with enhancement factors of [2, 3, 4, 5, 6, 7], JPEG compression at quality levels [10, 20, 30, 40, 50, 60], Gaussian noise addition with a

---

[2]https://huggingface.co/stabilityai/stable-diffusion-2-1-base
[3]https://huggingface.co/datasets/Gustavosta/Stable-Diffusion-Prompts

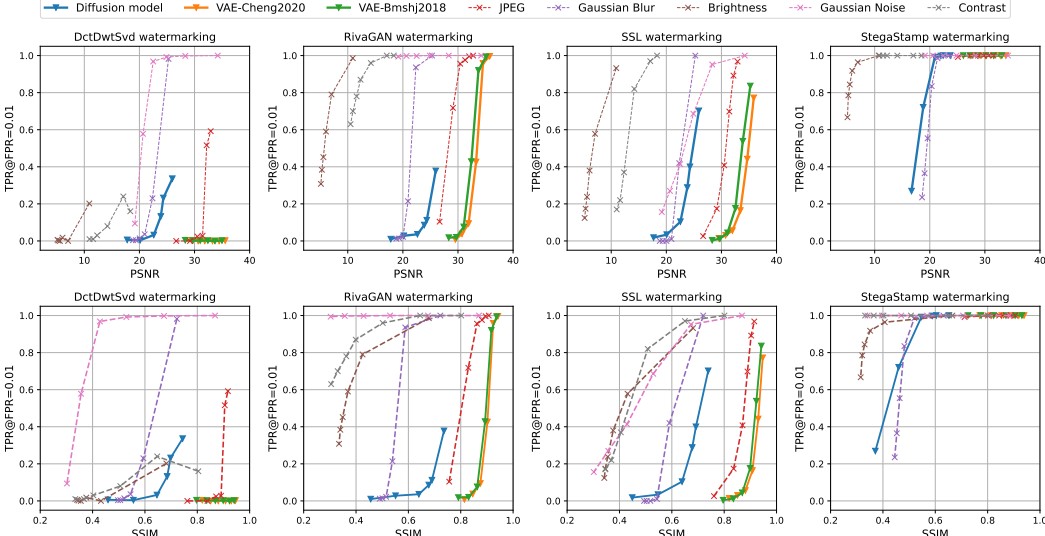

Figure 4: Quality-detectability tradeoff for four watermarking schemes under eight attack methods on the MS-COCO dataset. Regeneration attacks (Diffusion model, VAE-Cheng2020, and VAE-Bmshj2018) are highlighted for their performance. The x-axis shows image quality metrics (SSIM and PSNR, higher values indicate better quality), while the y-axis represents the detection metric True Positive Rate at 1% False Positive Rate (TPR@FPR=0.01, lower values are better for attackers). The strongest attacker should appear in the lower right corner of these plots. Regeneration attacks demonstrate superior performance compared to other attack methods, achieving both lower TPR and higher image quality. Quality-detectability tradeoff results for the SDP dataset are in Figure 7.

mean of 0 and standard deviations of [5, 10, 15, 20, 25, 30], and Gaussian blur with radii of [2, 4, 6, 8, 10, 12]. Further details are provided in Appendix B.

**Proposed attacks.** For regeneration attacks using variational autoencoders, we evaluate two pre-trained image compression models from the CompressAI library [5]: Bmshj2018 [3] and Cheng2020 [7]. Compression factors are set to [1, 2, 3, 4, 5, 6], where lower factors correspond to more heavily degraded images. For diffusion model attacks, we use the `Stable Diffusion-v2.1` model. The number of noise steps is set to [10, 30, 50, 100, 150, 200] (with $\sigma =$[0.10, 0.17, 0.23, 0.34, 0.46, 0.57]), and we employ pseudo numerical methods for diffusion models (PNDMs) [34] to generate samples. By adjusting the compression factors and noise steps, we achieve varying levels of perturbation for regeneration attacks.

**Evaluation metrics.** We evaluate the quality of attacked and watermarked images compared to the original cover image using two common metrics: Peak Signal-to-Noise Ratio (PSNR) defined as $\text{PSNR}(x, x') = -10 \cdot \log_{10}(\text{MSE}(x, x'))$, for images $x, x' \in [0, 1]^{c \times h \times w}$, and Structural Similarity Index (SSIM) [58] which measures perceptual similarity. To evaluate the diversity and quality of watermarked images, we use Fréchet Inception Distance (FID) [22] between the distributions of watermarked and unwatermarked images. To evaluate the robustness of the watermark, we compute the True Positive Rate (TPR) at a fixed False Positive Rate (FPR), specifically TPR@FPR=0.01. The detection threshold corresponding to FPR=0.01 is set according to each watermark's default configuration: correctly decoding 23/32, 59/96, and 32/48 bits for the respective watermarking methods, as described in the watermark settings.

## 5.1 Results and Analysis

This section presents detailed results and analysis of the regeneration attack experiments on different watermarking methods. Some attacking examples are shown in Figure 3.

Table 1: Performance of different watermarking methods. All methods successfully detect the embedded watermark.

| | MS-COCO Dataset | | | | SDP Generated Dataset | | | |
|---|---|---|---|---|---|---|---|---|
| Watermark | PSNR↑ | SSIM↑ | FID↓ | TPR@FPR=0.01↑ | PSNR↑ | SSIM↑ | FID↓ | TPR@FPR=0.01↑ |
| DwtDctSvd | 39.38 | 0.983 | 5.28 | 1.000 | 37.73 | 0.972 | 9.62 | 1.000 |
| RivaGAN | 40.55 | 0.978 | 10.83 | 1.000 | 40.64 | 0.979 | 13.56 | 1.000 |
| SSL | 41.79 | 0.984 | 18.86 | 1.000 | 41.88 | 0.983 | 23.87 | 1.000 |
| StegaStamp | 28.50 | 0.911 | 35.91 | 1.000 | 28.28 | 0.900 | 41.63 | 1.000 |

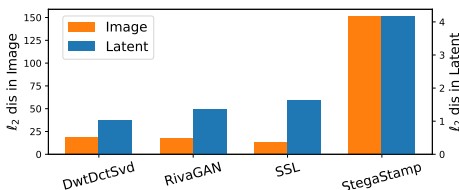

Figure 5: The $\ell_2$ distances between original images and watermarked ones. StegaStamp watermarked images are much more different in both pixel space and latent space.

**Watermarking performance without attacks.** Table 1 summarizes the watermarked image quality and detection rates for images watermarked by the four methods without any attacks. Each method—DwtDctSvd, RivaGAN, SSL, and StegaStamp—successfully embeds and retrieves messages from the images. These methods are post-processing techniques, adding watermarks to existing images. Among them, SSL achieves the highest PSNR and SSIM values, indicating superior perceptual quality and minimal visual distortion compared to the original images. DwtDctSvd achieves the lowest FID scores, suggesting the watermarked images maintain fidelity similar to clean images. In contrast, StegaStamp exhibits a noticeable drop in quality, with the lowest PSNR and highest FID scores. As illustrated in Figure 6, StegaStamp introduced noticeable blurring artifacts.

**Watermark removal effectiveness and image quality reservation.** Figure 4 and 7 present the quality-detectability tradeoff results from applying various regeneration attacks to remove watermarks. The VAE and diffusion model-based attacks (VAE-Bmshj2018, VAE-Cheng2020, Diffusion) consistently achieve over 99% removal rates for the DctDwtSvd, RivaGAN, and SSL watermarking methods, demonstrating their high effectiveness. In contrast, StegaStamp exhibits the highest robustness, with effective removal only achieved by the diffusion model with substantial noise. This resilience is partially attributed to StegaStamp's lower visual quality and higher perturbation levels (as indicated in Table 1, Figure 6 and Figure 5). Yet, increasing the noise level in diffusion models reduces StegaStamp's resistance. As demonstrated in Figure 4 and 7, higher noise levels improve the removal rate on StegaStamp, aligning with the theory that larger perturbations necessitate more noise for effective removal. This also aligns with the trade-off of lower image quality at higher noise levels. Overall, the consistently high removal rates across various watermarking schemes demonstrate the effectiveness of regeneration attacks for watermark removal. In terms of image quality preservation, the regeneration attacks generally maintain high quality, as indicated by PSNR and SSIM metrics. VAE models yield higher PSNR and SSIM scores, suggesting superior perceptual quality from a GAN-based perspective. However, qualitative inspection of example images in Figure 3 reveals the VAE outputs exhibit some blurring compared to the diffusion outputs. Since PSNR and SSIM are known to be insensitive to blurring artifacts [42, 58], we conclude that the choice of using regeneration with diffusion models should be guided by the specific requirements of each application.

**Potential defense.** Although we show that the proposed attack is guaranteed to remove any pixel-based invisible watermarks, it is not impossible to detect AI-generated images. Semantic watermarks offer a viable detection method, with further details and results discussed in Appendix E.

## 6 Conclusion

We proposed a regeneration attack on invisible watermarks that combines destructive and constructive attacks. Our theoretical analysis proved that the proposed regeneration attack is able to remove certain invisible watermarks from images and make the watermark undetectable by any detection algorithm. We showed with extensive experiments that the proposed attack performed well empirically. The proofs and experiments revealed the vulnerability of invisible watermarks. Given this vulnerability, we explored an alternative defense that uses visible but semantically similar watermarks. Our findings on the vulnerability of invisible watermarks underscore the need for shifting the research/industry emphasis from invisible watermarks to their alternatives.

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

# A  Additional Discussion in FAQ style

**How do we control a certain degree of Certified Watermark Freeness apriori?** We note that our guarantee in Theorem 4.3 depends on the specific watermark injected on specific image instance through the unknown local Lipschitz parameter $L_{x,w}$. Specifying a fixed CWF level requires us to have a uniform upper bound of $L_{x,w}$ independent to $x$ and $w$. To give two concrete examples, when the embedding $\phi$ is chosen trivial to be the identity map, then we can take $L_{x,w} \leq 1$. When $\phi$ is a lower pass filtering, e.g., Fourier transform than removing all high-frequency components except the top $k$ dimension, then we can bound $L_{x,w} \leq \sqrt{k}/n$ where $n$ is the number of pixels. More generally, any linear transformation with bounded operator norm works.

**Is there a uniform bound for neural embedding $\phi$ such as those from VAE and diffusion models?** We believe there is, but these bounds might be too conservative to use in practice. In particular, they might be $\gg 1$ due to the widely observed adversarial examples [52, 19]. If the injected watermarks are carefully designed such that the injected perturbation is aligned with an adversarial perturbation, then the resulting watermarked images will be more resilient to our attacks that leverage the neural embeddings. They will not be more resilient to our attacks that do not use neural embeddings though. In practice, we found that in all existing watermarks, neural embeddings result in substantially smaller distortions to the original images while suppressing $L_{x,w}$ substantially (Figure 5).

**Other ways of achieving CWF without adding Gaussian noise.** Research from the golden age of digital watermarking [39, 9, 16, 8] has proposed methods such as quantization for removing watermarks that provide similar tradeoffs in practice to the watermarks that we experimented with in this paper. However, these methods, in general, do not come with provable CWF guarantees. It is critical that the watermark removal attack is randomized to enjoy similar properties to what we have in Theorem 4.3. That said, the classical removal attacks can be used in constructing better embedding function $\phi$ which may help reducing the local Lipschitz parameter and, thus, improving the tradeoff between certified removal and utility of the regenerated image. Besides Gaussian noise, we can also add Laplace noise and other well-known perturbation mechanisms from the differential privacy and cryptography research community. Thoroughly investigating the impact of the choice of noise (and randomized quantization approaches) is an interesting direction of future research.

# B  Additional Method Details

## B.1  Invisible Watermarking Methods

In this section, we review several well-established invisible watermarking schemes that are evaluated in our experiments (Section 5). These approaches cover a range of methods, including traditional signal processing techniques and more recent deep learning methods. They include the default watermarking schemes employed by the widely used Stable Diffusion models [47]. We show some invisible watermarking examples in Figure 6.

**DwtDctSvd watermarking.** The DwtDctSvd watermarking method [41] combines Discrete Wavelet Transform (DWT), Discrete Cosine Transform (DCT), and Singular Value Decomposition (SVD) to embed watermarks in color images. First, the RGB color space of the cover image is converted to YUV. DWT is then applied to the Y channel, and DCT divides it into blocks. SVD is performed on each block. Finally, the watermark is embedded into the blocks. DwtDctSvd is the default watermark used by Stable Diffusion.

**RivaGAN watermarking.** RivaGAN [67] presents a robust image watermarking method using GANs. It employs two adversarial networks to assess watermarked image quality and remove watermarks. An encoder embeds the watermark, while a decoder extracts it. By combining these, RivaGAN offers superior performance and robustness. RivaGAN is another watermark used by Stable Diffusion.

**StegaStamp watermarking.** StegaStamp [53] is a robust CNN-based watermarking method. It uses differentiable image perturbations during training to improve noise resistance. Additionally, it incorporates a spatial transformer network to resist minor perspective and geometric changes like cropping. This adversarial training and spatial transformer enable StegaStamp to withstand various attacks.

| Original Image | DwtDctSvd WM | RivaGAN WM | SSL WM | StegaStamp WM |
| --- | --- | --- | --- | --- |

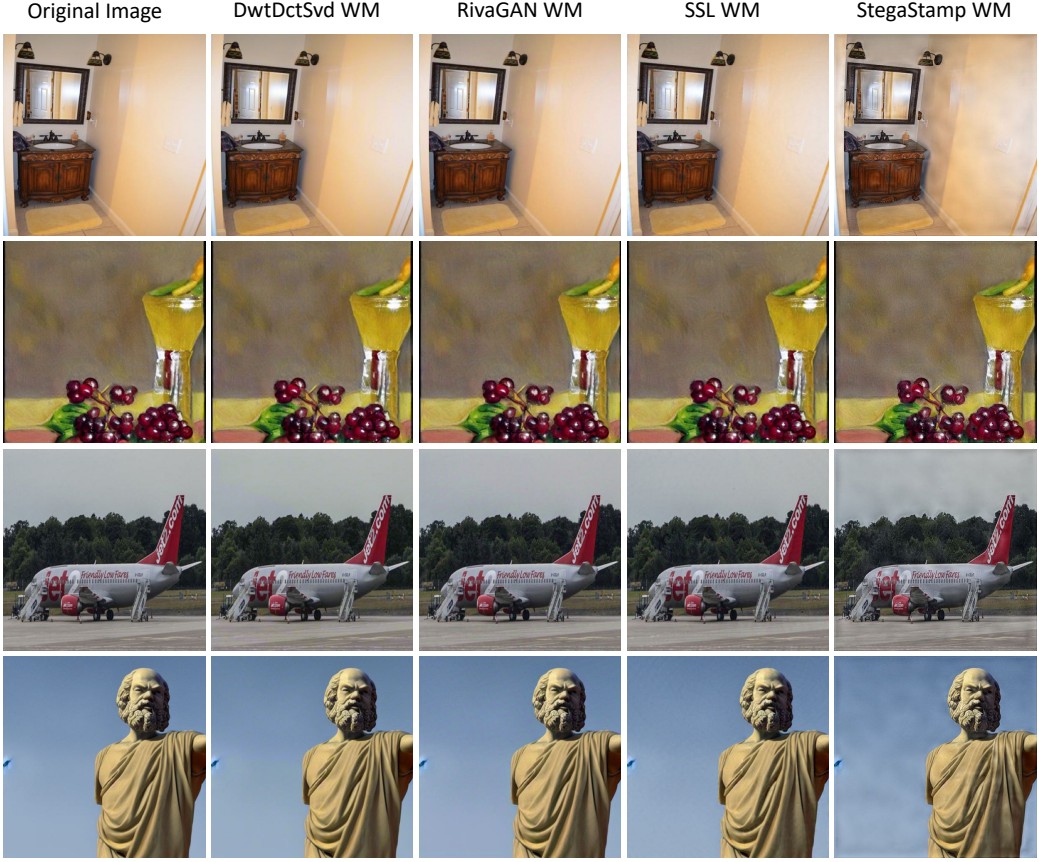

Figure 6: More examples of different invisible watermarking methods.

**SSL watermarking.** SSL watermarking [15] utilizes pre-trained neural networks' latent spaces to encode watermarks. Networks pretrained with self-supervised learning (SSL) extract effective features for watermarking. The method embeds watermarks through backpropagation and data augmentation, then detects and decodes them from the watermarked image or its features.

## B.2    Existing Attacking Methods

In this section, we review common attacking methods that aim to degrade or remove invisible watermarks in images. These methods are widely used to measure the robustness of watermarking algorithms against removal or tampering [67, 53, 15, 14, 60]. The attacking methods can be categorized into destructive attacks, where the watermark is considered part of the image and actively removed by corrupting the image, and constructive attacks, where image processing techniques like denoising are used to obscure the watermark.

Destructive attacks intentionally corrupt the image to degrade or remove the embedded watermark. Common destructive attack techniques include:

**Brightness/Contrast adjustment.** This attack adjusts the brightness and contrast parameters of the image globally. Adjusting the brightness/contrast makes the watermark harder to detect.

**JPEG compression.** JPEG is a common lossy image compression technique. It has a quality factor parameter that controls the amount of compression. A lower quality factor leads to more loss of fine details and a higher chance of degrading the watermark.

**Gaussian noise.** This attack adds random Gaussian noise to each pixel of the watermarked image. The variance of the Gaussian noise distribution controls the strength of the noise. A higher variance leads to more degradation of the watermark signal.

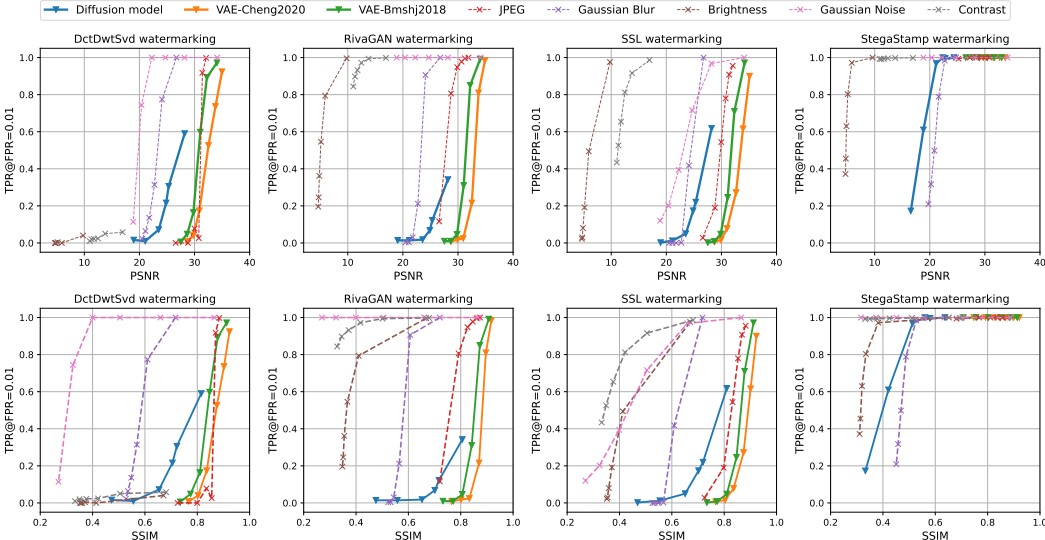

Figure 7: Trade-off between quality and detectability for four watermarking schemes tested against eight attack methods on the Stable Diffusion Prompt (SDP) dataset.

Constructive attacks aim to remove the watermark by improving image quality and restoring the original unwatermarked image. Examples include:

**Gaussian blur.** Blurring the image by convolving it with a Gaussian kernel smoothens the watermark signal and makes it less detectable. The kernel size and standard deviation parameter control the level of blurring.

Other recent attacking methods [32, 49, 37, 1] aim to remove invisible image watermarks using generative models. Concurrent to our work, [32] use propose DiffWA, a conditional diffusion model with distance guidance for watermark removal. However, DiffWA only works for low-resolution images and lacks theoretical guarantees. [49] train a substitute classifier and conduct projected gradient descent (PGD) attacks on it to deceive black-box watermark detectors. However, their approach requires multiple queries to the target generator. [37] propose an adaptive attack that locally replicates secret watermarking keys by creating differentiable surrogate keys used to optimize the attack parameters. However, they assume the attacker knows the watermarking method, which is a stronger assumption than ours. Our method removes invisible watermarks from high-resolution images, provides theoretical justifications, and does not assume knowledge of the watermarking algorithm.

## C   Remarks on Our Theoretical Guarantee

In this section, we discuss more about the theoretical guarantees in Section 4. For the readers' convenience, we first restate the definitions and theorems and then discuss their implications and interpretation. The proofs can be found in Appendix D.

### C.1   Certified Watermark Removal

How do we quantify the ability of an attack algorithm to remove watermarks? We argue that if after the attack, no algorithm is able to distinguish whether the result is coming from a watermarked image or the corresponding original image without the watermark, then we consider the watermark certifiably removed. More formally:

**Definition C.1** ($f$-Certified-Watermark-Free)**.** We say that a watermark removal attack is $f$-Certified-Watermark-Free (or $f$-CWF) against a watermark scheme for a non-increasing function $f : [0, 1] \to [0, 1]$, if for any detection algorithm $\mathsf{Detect} : \mathcal{X} \times \mathsf{aux} \to \{0, 1\}$, the Type II error (false negative rate) $\epsilon_2$ of $\mathsf{Detect}$ obeys that $\epsilon_2 \geq f(\epsilon_1)$ for all Type I error $0 \leq \epsilon_1 \leq 1$.

| Original Image | DwtDctSvd WM | Brightness | JPEG | Gaussian blur | **VAE attack** | **Diffusion attack** |
|---|---|---|---|---|---|---|

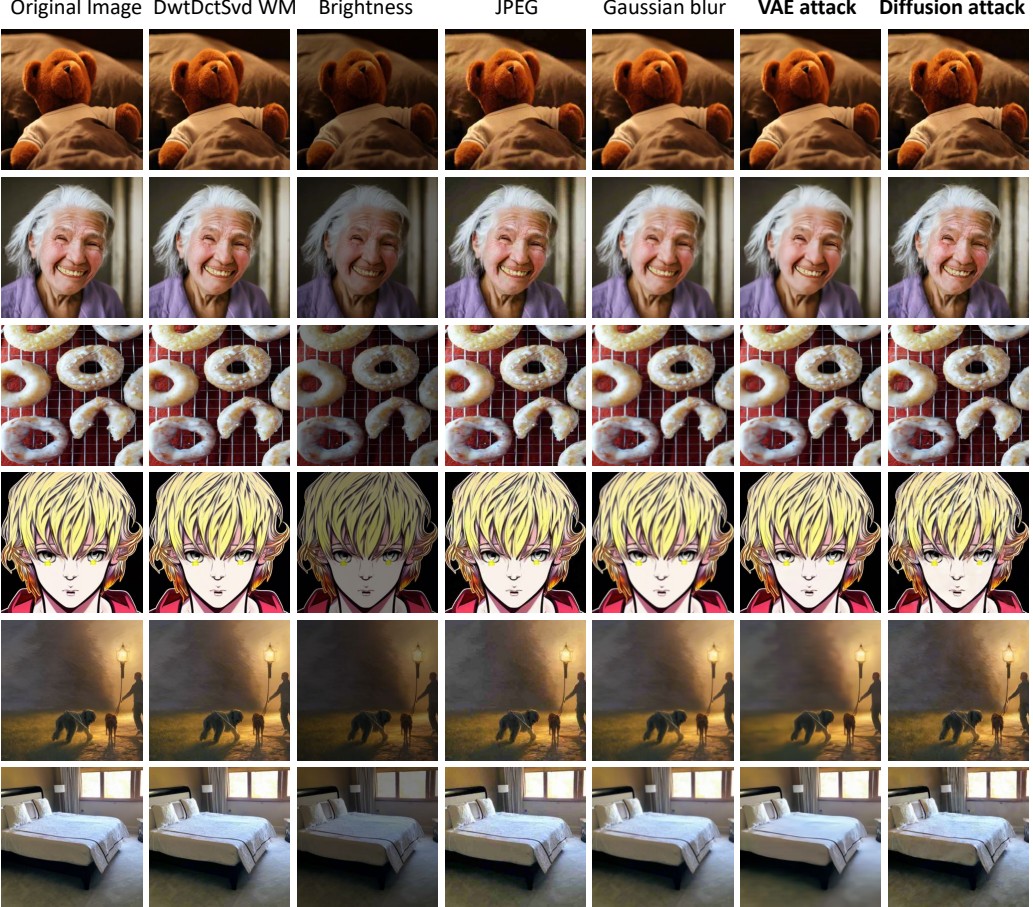

Figure 8: More examples of watermarking attacks against DwtDctSvd.

Let us also define a parameter to quantify the effect of the embedding function $\phi$.

**Definition C.2** (Local Watermark-Specific Lipschitz property). We say that an embedding function $\phi : \mathcal{X} \to \mathbb{R}^d$ satisfies $L_{x,w}$-Local Watermark-Specific Lipschitz property if for a watermark scheme $w$ that generates $x_w$ with $x$,

$$\|\phi(x_w) - \phi(x)\| \leq L_{x,w}\|x_w - x\|.$$

The parameter $L_{x,w}$ measures how much the embedding compresses the watermark added on a particular clean image $x$. If $\phi$ is identity, then $L_{x,w} \equiv 1$. If $\phi$ is a projection matrix to a linear subspace then $0 \leq L_{x,w} \leq 1$ depending on the magnitude of the component of $x_w - x$ in this subspace. For a neural image embedding $\phi$, the exact value of $L_{x,w}$ is unknown but given each $x_w$ and $x$ it can be computed efficiently.

**Theorem C.3.** *For a $\Delta$-invisible watermarking scheme with respect to $\ell_2$-distance. Assume the embedding function $\phi$ of the diffusion model is $L_{x,w}$-Locally Lipschitz. The randomized algorithm $\mathcal{A}(\phi(\cdot) + \mathcal{N}(0, \sigma^2 I_d))$ produces a reconstructed image $\hat{x}$ which satisfies $f$-CWF with*

$$f(\epsilon_1) = \Phi\left(\Phi^{-1}(1 - \epsilon_1) - \frac{L_{x,w}\Delta}{\sigma}\right),$$

*where $\Phi$ is the Cumulative Density Function function of the standard normal distribution.*

Figure 2 illustrates what the tradeoff function looks like. The result says that after the regeneration attack, it is impossible for any detection algorithm to correctly detect the watermark with high confidence. In addition, it shows that such detection is as hard as telling the origin of a single sample $Y$ from either of the two Gaussian distributions $\mathcal{N}(0, 1)$ and $\mathcal{N}(L_{x,w}\Delta/\sigma, 1)$.

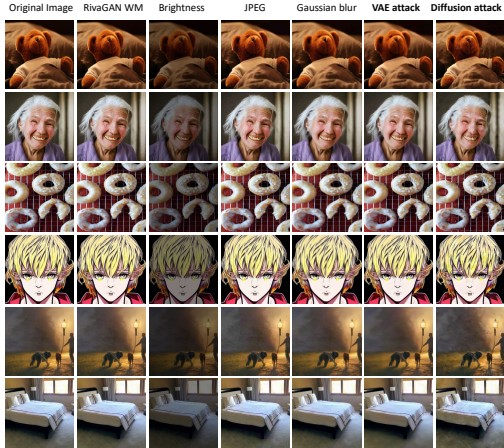

Figure 9: Examples of different attacks against RivaGAN watermark.

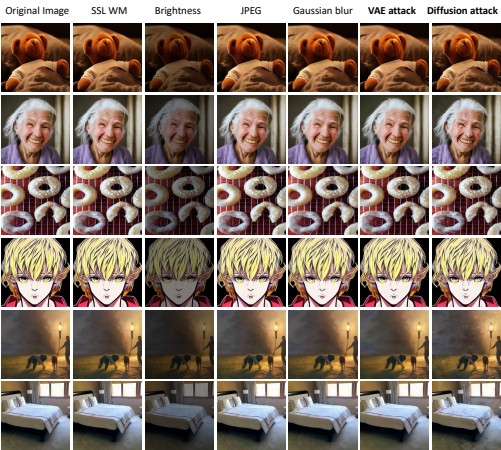

Figure 10: Examples of different attacks SSL watermark.

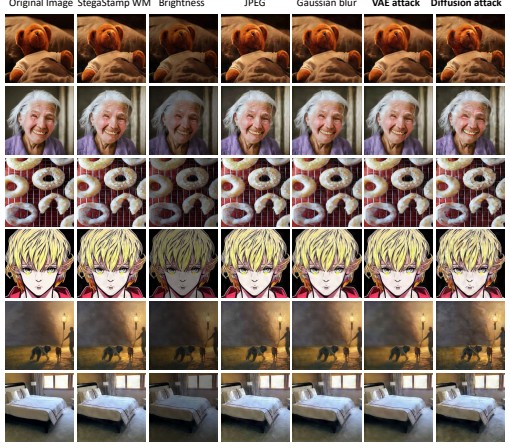

Figure 11: Examples of different attacks against StegaStamp watermark.

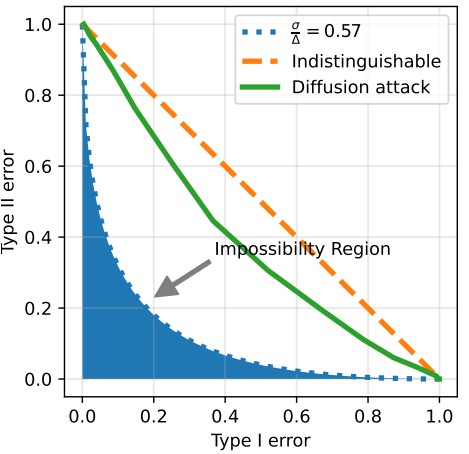

Figure 12: Theoretical and empirical trade-off functions for RivaGAN watermark.

The proof in Section D leverages an interesting connection to a modern treatment of differential privacy [13] known as the Gaussian differential privacy [12]. The work of [12] itself is a refinement and generalization of the pioneering work of [59] and [28] which established a tradeoff-function view.

**Discussion.** Let us instantiate the algorithm with a latent diffusion model by choosing $\sigma = \sqrt{(1 - \alpha(t^*))/\alpha(t^*)}$ (see Algorithm 1) and discuss the parameter choices.

*Remark* C.4 (Two trivial cases). Observe that when $\alpha(t^*) = 0$, the result of the reconstruction does not depend on the input $x_w$, thus there is no information about the watermark in $\hat{x}(0)$, i.e., the trade-off function is $f(\epsilon_1) = 1 - \epsilon_2$ — perfectly watermark-free, however, the information about $x$ (through $x_w$) is also lost. When $\alpha(t^*) = 1$, the attack trivially returns $\hat{x}(0) = x_w$, which does not change the performance of the original watermark detection algorithm at all (and it could be perfect, i.e., $\epsilon_1 = \epsilon_2 = 0$).

*Remark* C.5 (Choice of $t^*$). In practice, the best choice $t^*$ is in between the two trivial cases, i.e., one should choose it such that $L_{x,w}\Delta\sqrt{\alpha(t^*)/(1 - \alpha(t^*))}$ is a small constant. The smaller the constant, the more thoroughly the watermark is removed. The larger the constant, the higher the fidelity of the regenerated image w.r.t. $x_w$ (thus $x_0$ too).

*Remark* C.6 (VAE). Strictly speaking, Theorem 4.3 does not directly apply to VAE because the noise added on the latent embedding depends on the input data. So if it chooses $\sigma(x_0) = 1$ and

$\sigma(x_w) = 0$, then it is easy to distinguish between the two distributions. We can still provide provable guarantees for VAE if either the input image is artificially perturbed (so VAE becomes a denoising algorithm) or the latent space is artificially perturbed after getting the embedding vector. When $\sigma(x)$ itself could be stable, more advanced techniques from differential privacy such as Smooth Sensitivity or Propose-Test-Release can be used to provide certified removal guarantees for the VAE attack. In practice, we find that the VAE attack is very effective in removing watermarks *as is* without adding additional noise.

*Remark* C.7 (The role of embedding function $\phi$). Readers may wonder why having an embedding function $\phi$ is helpful for removing watermarks. We give three illustrative examples.

**Pixel quantization.** This $\phi$ is effective against classical Least Significant Bit (LSB) watermarks. By removing the lower-significance bits $\phi(x) = \phi(x_w)$ thus $L_{x,w} = 0$.

**Low-pass filtering.** By choosing $\phi$ to be a low-pass filter, one can effectively remove or attenuate watermarks injected in the high-frequency spectrum of the Fourier domain, hence resulting in a $L_{x,w} \ll 1$ for these watermarks.

**Deep-learning-based image embedding.** Modern deep-learning-based image models effectively encode a "natural image manifold," which allows a natural image $x$ to pass through while making the added artificial watermark $\epsilon$ smaller. To be more concrete, consider $\phi$ to be a linear projection to a $d$-dimensional "natural image subspace". For a natural image $x$ and watermarked image $x_w = x + \varepsilon$, we have $\phi(x_w) = \phi(x) + \phi(\varepsilon) = x + \phi(\varepsilon)$. If $\varepsilon \sim \mathcal{N}(0, \sigma_w^2 I_n)$ then $\mathbb{E}[\|\phi(\varepsilon)\|^2] = d\sigma_w^2 \ll n\sigma_w^2 = \mathbb{E}[\|\varepsilon\|^2]$. This projection compresses the magnitude of the watermark substantially while preserving the *signal*, thereby boosting the effect of the noise added in the embedding space in obfuscating the differences between watermarked and unwatermarked images.

Finally, we note that while Theorem 4.3 and 4.4 are specific to $\ell_2$-distance, the general idea applies to other distance functions (e.g., $\ell_1$ distance). $\ell_2$-distance is natural for the Gaussian noise natively introduced by diffusion and VAE-based regeneration attacks.

## C.2    Utility Guarantees

In this section, we prove that the regenerated image $\hat{x}$ is close to the original (unwatermarked) image $x_0$. This is challenging because the denoising algorithm only gets access to the noisy version of the watermarked image.

Interestingly, we can obtain a general extension lemma showing that for any black-box generative model that can successfully denoise a noisy yet unwatermarked image with high probability, the same result also applies to the watermarked counterpart, except that the failure probability is slightly larger.

**Theorem C.8.** *Let $x_0$ be an image with $n$ pixels and $\phi : \mathbb{R}^n \to \mathbb{R}^d$ be an embedding function. Let $\mathcal{A}$ be an image generation / denoising algorithm such that with probability at least $1 - \delta$, $\|\mathcal{A}(\phi(x_0) + \mathcal{N}(0, \sigma^2 I_d)) - x_0\| \leq \xi_{x_0,\sigma,\delta}$. Then for any $\Delta$-invisible watermarking scheme that produces $x_w$ from a clean image $x_0$, then $\hat{x} = \mathcal{A}(\phi(x_w) + \mathcal{N}(0, \sigma^2 I_d))$ satisfies that*

$$\|\hat{x} - x_0\| \leq \xi_{x_0,\sigma,\delta}$$

*with a probability at least $1 - \tilde{\delta}$ where*

$$\tilde{\delta} = \min_{v \in \mathbb{R}} \left\{ \delta \cdot e^v + \Phi\left(\frac{\tilde{\Delta}}{2\sigma} - \frac{v\sigma}{\tilde{\Delta}}\right) - e^v \Phi\left(-\frac{\tilde{\Delta}}{2\sigma} - \frac{v\sigma}{\tilde{\Delta}}\right) \right\}$$

*in which $\Phi$ denotes the standard normal CDF and $\tilde{\Delta} := L_{x_0,w}\Delta$.*

The theorem says that if a generative model is able to denoise a noisy version of the original image, then the corresponding watermark-removal attack using this generative model provably produces an image with similar quality.

**Corollary C.9.** *The expression for $\tilde{\delta}$ above can be (conservatively) simplified to*

$$\tilde{\delta} \leq e^{\frac{L_{x,w}^2 \Delta^2}{\sigma^2}} \cdot \delta^{1/2}.$$

*For example, if $\sigma \asymp L_{x,w}\Delta$, then this is saying that if $\xi_{x_0,\sigma,\delta}$ depends logarithmically on $1/\delta$, the same exponential tail holds for denoising the watermarked image.*

The above result is powerful in that it makes no assumption about what perturbation the watermarking schemes could inject and which image generation algorithm we use. We give a few examples below.

For denoising algorithms with theoretical guarantees, e.g., TV-denoising [27, Theorem 2], our results imply provable guarantees on the utility for the watermark removal attack of the form, "w.h.p., $\frac{1}{n}\|\hat{x} - x_0\|^2 = \tilde{O}\left(\frac{\sigma \mathrm{TV}_{2d}(x_0)}{n}\right)$", i.e., vanishing mean square error (MSE) as $n$ gets bigger.

For modern deep learning-based image denoising and generation algorithms where worst-case guarantees are usually intractable, Theorem 4.4 is still applicable for each image separately. That is to say, as long as their empirical denoising quality is good on an unwatermarked image, the quality should also be good on its watermarked counterpart.

## D    Proofs of Technical Results

*Proof of Theorem 4.3.* By the conditions on the invisible watermark and the local Lipschitz assumption on $\phi$, we get that
$$\|\phi(x_0) - \phi(x_w)\|_2 \le L_{x_0,w}\Delta.$$
This can be viewed as the $\ell_2$-local-sensitivity of $\phi$ at $x_0$ in the language of differential privacy literature.

The two candidate distributions are $\mathcal{N}(\phi(x_0), \sigma^2 I)$ and $\mathcal{N}(\phi(x_w), \sigma^2 I)$. Let $T(P, Q) : [0, 1] \to [0, 1]$ be the tradeoff function for distinguishing between distributions $P$ and $Q$ as in Definition 2.1 of [12]. The tradeoff function outputs the Type II error of the likelihood ratio test of the two point hypotheses as a function of the Type I error. By the Neyman-Pearson lemma, the likelihood ratio test is uniform most powerful, thus the tradeoff function provides a lower bound of any test in distinguishing $P$ and $Q$.

First notice that translation and scaling by any non-zero constant (a non-zero linear transformation) does not change the tradeoff function, thus

$$T\left(\mathcal{N}(\phi(x_0), \sigma^2 I), \mathcal{N}(\phi(x_w), \sigma^2 I)\right) = T\left(\mathcal{N}(0, I), \mathcal{N}(\frac{\phi(x_w) - \phi(x_0)}{\sigma^2}, I)\right)$$

Next, observe that the likelihood ratio of the two multivariate isotropic normal distributions remains the same when we collapse it to the 1-dimension along the vector $\phi(x_0) - \phi(x_w)$, i.e.,

$$T\left(\mathcal{N}(0, I), \mathcal{N}(\frac{\phi(x_w) - \phi(x_0)}{\sigma^2}, I)\right) = T\left(\mathcal{N}(0, 1), \mathcal{N}(\frac{\|\phi(x_w) - \phi(x_0)\|}{\sigma^2}, 1)\right). \tag{5}$$

Lemma 2.9 of [12] states that for any post-processing function $h$,

$$T(h(P), h(Q)) \ge T(P, Q) \tag{6}$$

which is known to information theorists as the information processing inequality of the Hockey-Stick divergence.

Consider a particular randomized post-processing function $h$ that first adds $\mathcal{N}(0, v)$ then divides $\sqrt{1 + v}$.

$$T\left(\mathcal{N}(0, 1), \mathcal{N}(\frac{L_{x_0,w}\Delta}{\sigma^2}, 1)\right) \le T\left(\mathcal{N}(0, 1), \mathcal{N}(\frac{L_{x_0,w}\Delta}{\sigma^2(\sqrt{1+v})}, 1)\right) \underset{\substack{\uparrow \\ \text{for a specific } v}}{=} (5)$$

To see the last identity, observe that we can choose $v > 0$ such that $\frac{L_{x_0,w}\Delta}{\sigma^2(\sqrt{1+v})} = \frac{\|\phi(x_w) - \phi(x_0)\|}{\sigma^2}$, thanks to the local Lipschitz assumption. To put things together, we get

$$T\left(\mathcal{N}(0, 1), \mathcal{N}(\frac{L_{x_0,w}\Delta}{\sigma^2}, 1)\right) \le T\left(\mathcal{N}(\phi(x_0), \sigma^2 I), \mathcal{N}(\phi(x_w), \sigma^2 I)\right)$$

The left hand side is the tradeoff function of two univariate Gaussian with variance 1, by Equation (6) of [12], the tradeoff function can be written as $\Phi\left(\Phi^{-1}(1 - \alpha) - \frac{L_{x_0,w}\Delta}{\sigma^2}\right)$ where $\Phi$ is the cumulative density function of the standard Gaussian distribution.

Finally, by applying the the postprocessing property (6) again by instantiating $h$ to be the re-generation procedure, we get that the regenerated image $\hat{x}$ also satisfies the same tradeoff function as stated above. $\qquad\square$

*Proof of Theorem 4.4.* The key idea is to use the definition of indistinguishability (differential privacy, but for a fixed pair of neighbors, rather than for all neighbors). So we say two input $x, x'$ are $(v, w)$-indistinguishable using the output of a mechanism $\mathcal{M}$ if for any event $S$,

$$\Pr[\mathcal{M}(x) \in S] \le e^v \Pr[\mathcal{M}(x') \in S] + w.$$

and the same also true when $x, x'$ are swapped. In our case, we have already shown that (from the proof of Theorem 4.3) for any post-processing algorithm $\mathcal{A}$, $x_0$ and $x_w$ are indistinguishable using $\hat{x}$ in the trade-off function sense. [2] obtained a "dual" characterization which says that the same Gaussian mechanism satisfies $(v, w)$-indistinguishability with

$$w = \Phi\left(\frac{\tilde{\Delta}}{2\sigma} - \frac{v\sigma}{\tilde{\Delta}}\right) - e^v \Phi\left(-\frac{\tilde{\Delta}}{2\sigma} - \frac{v\sigma}{\tilde{\Delta}}\right)$$

for all $v \in \mathbb{R}$. By instantiating the event $S$ to be that $\|\hat{x} - x_0\| > \xi_{x_0, \sigma, \delta}$, then we get

$$\Pr_{x_w}[\|\hat{x} - x_0\| > \xi_{x_0, \sigma, \delta}] \le e^v \Pr_{x_0}[\|\hat{x} - x_0\| > \xi_{x_0, \sigma, \delta}] + w$$
$$= e^v \cdot \delta + w.$$

This completes the proof. $\qquad\square$

*Proof of Corollary 4.5.* The $w, v$ "privacy profile" implies a Renyi-divergence bound (one can also get that directly from the Renyi-DP of gaussian mechanism) which implies (by Proposition 10 of [40]) that

$$\tilde{\delta} \le \min_{u \ge 1}\left(e^{\frac{u\tilde{\Delta}^2}{2\sigma^2}} \cdot \delta\right)^{(u-1)/u}.$$

The stated result is obtained by setting $u = 2$. $\qquad\square$

## E  Defense with Semantic Watermarks

In this section, we discuss possible defenses that are resilient to the proposed attack, and although Theorem 4.3 has guaranteed that no detection algorithm will be able to detect the watermark after our attack, the guarantee is based on the invisibility with respect to $\ell_2$ distance. Therefore, by relaxing that invisibility constraint and thus making the watermark more visible, we may be able to prevent the watermark from being removed. One less-harmful way to loosen the invisibility constraint is with semantic watermarks. As shown in Figure 13, pixel-based watermarks such as DwtDctSvd keep the image almost intact, while semantic watermarks change the image significantly but retain its content.

### E.1  Tree-Ring Watermarks

Tree-Ring Watermarking [60] is a new technique that robustly fingerprints diffusion model outputs in a way that is semantically hidden in the image space. An image with a Tree-Ring watermark does not look the same as the image, but it is semantically similar (in Figure 13, both the original and the semantically watermarked image contain an astronaut riding a horse in Zion National Park). Unlike existing methods that perform post-hoc modifications to images after sampling, Tree-Ring Watermarking subtly influences the entire sampling process, resulting in a model fingerprint. The watermark embeds a pattern into the initial noise vector used for sampling. These patterns are structured in Fourier space so that they are invariant to convolutions, crops, dilations, flips, and rotations. After image generation, the watermark signal is detected by inverting the diffusion process to retrieve the noise vector, which is then checked for the embedded signal. [60] demonstrated that Tree-Ring Watermarking can be easily applied to arbitrary diffusion models, including text-conditioned Stable Diffusion, as a plug-in with negligible loss in FID.

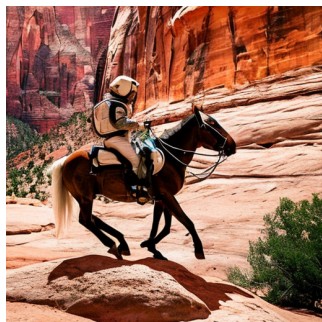 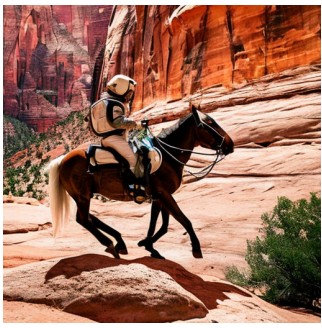 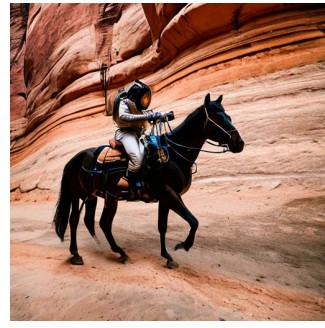

(a) No Watermark       (b) Pixel WM       (c) Semantic WM

Figure 13: The image with a pixel-based watermark such as DwtDctSvd looks almost the same as the original. The image with a semantic watermark such as Tree-Ring contains the same content but is visibly different from the original. Original image generated with the prompt "an astronaut riding a horse in Zion National Park" from [60].

Table 2: Tree-Ring watermarks are robust against the regeneration attacks.

| Attacker | MS-COCO TPR@FPR=0.01↓ | SDP Generated TPR@FPR=0.01↓ |
|---|---|---|
| Brightness-2 | 1.000 | 1.000 |
| Contrast-2 | 1.000 | 1.000 |
| JPEG-50 | 1.000 | 0.994 |
| Gaussian noise-5 | 1.000 | 0.996 |
| Gaussian blur-6 | 1.000 | 1.000 |
| VAE-Bmshj2018-3 | 0.998 | 0.994 |
| VAE-Cheng2020-3 | 1.000 | 0.994 |
| Diffusion model-60 | 1.000 | 0.998 |

## E.2  Defense Experiments

To evaluate Tree-Ring as an alternative watermark, we use the same datasets from the previous experiments in Section 5 - MS-COCO and SDP. However, since Tree-Ring adds watermarks during the generation process of diffusion, it cannot directly operate on AI-generated images. Instead, it needs textual inputs that describe the content of the images. We use captions from MS-COCO and the user prompts of SDP datasets as the input prompts. The selected set of attacks (including our proposed attack) is applied to the Tree-Ring watermarked images.

As shown in Table 2, Tree-Ring watermarks show exceptional robustness against all the attacks tested. However, such robustness does not come for free. We depict the $\ell_2$-distances between original images and watermarked ones in Figure 14. Images with Tree-Ring watermarks are significantly more different from the original images in both the pixel space and the latent space. These results indicate that Tree-Ring, as an instance of semantic watermarks, shows the potential to be an alternative solution to the image watermarking problem. However, as our theory predicts, the robustness comes at the price of more visible differences.

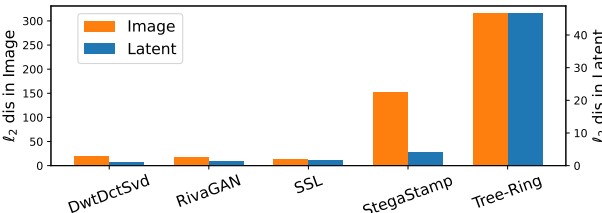

Figure 14: The $\ell_2$ distances between original images and watermarked ones. Tree-Ring watermarked images are much more different in both pixel space and latent space, making Tree-Ring a *visible* watermark.

