# OpenReview forum: "Invisible Image Watermarks Are Provably Removable Using Generative AI"
_NeurIPS.cc/2024/Conference — NeurIPS 2024 poster_

### Official Review · Reviewer_ii78 · 2024-06-23

**Soundness:** 4
**Presentation:** 3
**Contribution:** 3
**Rating:** 7
**Confidence:** 3

**Summary:**

This paper investigates the resilience of invisible watermarks embedded in images against removal attacks. The authors propose a new class of attacks, called regeneration attacks, which combine adding random noise to the image and then reconstructing the image using generative models. The study demonstrates that these attacks can effectively remove invisible watermarks, including the resilient RivaGAN, while maintaining image quality.

**Strengths:**

Introduces a new Image Watermarks attack leveraging generative models, providing a fresh perspective on watermark removal.

Offers formal proofs to demonstrate the effectiveness of the proposed attacks.

Provides extensive empirical results showing the success of the attacks across different watermarking methods.

The writing is clear and well-organized.

**Weaknesses:**

While invisible watermarks are highly vulnerable, semantic watermarks are less affected by the proposed attacks.

**Questions:**

N/A

---

> ### Author Rebuttal · Authors · 2024-08-06
>
> We greatly appreciate your positive feedback and the opportunity to address your concern.
>
> > "While invisible watermarks are highly vulnerable, semantic watermarks are less affected by the proposed attacks."
>
> The primary purpose of this paper is not to propose an attack that can remove any type of watermark. Our goal is to raise awareness about the vulnerability of all invisible watermarks to some extent. We believe we have successfully demonstrated this point and provided valuable insights into alternative solutions in light of these vulnerabilities. Therefore, this additional information should not be viewed as a weakness but rather as a significant contribution to the community. Our attack is important to motivate the community to study semantic watermarks.
>
> Thank you again for your feedback, and we are looking forward to your stronger support!

---

> > ### Comment · Reviewer_ii78 · 2024-08-09
> >
> > Thank you for your comments. I will keep my positive score.

---

### Official Review · Reviewer_V7G3 · 2024-06-24

**Soundness:** 3
**Presentation:** 3
**Contribution:** 3
**Rating:** 7
**Confidence:** 5

**Summary:**

The main idea proposed in this paper is that regenerating images using other (pretrained) generative AI (e.g., vae, diffusion models) can provably remove any invisible watermarks embedded in a given image. Accompanying this, the paper can be divided into the following parts: (1) intro of the proposed regeneration methods; (2) proof of the removal guarantee (by any regeneration methods falls into the definition of this paper, not limited to vae or diffusion; and (3) empirical experiments to support their claim "the regeneration methods is effective, especially using defussion models."

**Strengths:**

The idea proposed in this work original.
The paper is clearly written. The use of math does not create burden in readability but helps to understand the main idea.

**Weaknesses:**

The weakness is listed below in order, from major to minor:

Weakness 1 (Major): Experiment settings and evaluation are not rigorous.

The key experiment and result to support the authors claim of the effectiveness of their method will be those that can show the “strength” of their proposed attacks, but the current evaluation experiment is poorly designed. Reporting the watermark detection acc. (as in Table 2) using a fixed attack point (e.g., JPEG 50, or the selected VAE’s, etc., where they all have tunable parameters) cannot faithfully show the strength. This make the PSNR value reported in Table 2 meaningless. Instead, if there is a tradeoff between the image quality and the watermark detectability expected, the authors should report the maximal PSNR (the best image quality) for each attack method when the same detection ability (e.g., fail the watermark detection) of the decoder is achieved (e.g., with the same bitwise acc.) by tuning the hyperparameters of the attack methods (e.g., the JPEG quality factor, the compression index of the selected VAE model used in this paper, the noise level of the diffusion regeneration as Fig. 5, etc.). In this regard, the attack method with the best image quality can be argued as the strongest reasonably. Alternatively, the authors may consider profiling the quality and detectability tradeoff as proposed in [1]. The current evaluation experiment is not sufficient to support the claim made in line 71-73.

Weakness 2 (Major): Conclusion with insufficient support.

In line 13-14 “Our findings underscores … to semantic-preserving watermarks” is not convincing. For example, let’s say we want to prevent misusing generated images. In Fig 6., the authors visualize the StegaStamp watermark, where there are obvious abnormal patterns embedded into the image (in another word, they are not `invisible’ and humans can tell these images are suspicious). I would suspect that the attacked images of StegaStamp after applying the methods proposed by the authors will still have visible artifacts that are enough to raise human caution and suspect they are not original images—so it may be reasonable to argue that StegaStamp can successfully prevent image misuse and do not need a shift as the authors claimed. My suspicions are raised by the following results presented by the authors: (i) The visualizations of the attack methods proposed by the authors (e.g., Diffusion Attack in Fig 3. And 7.) are on DwtDctSVD watermarks and they already contain visible artifacts; (ii) corresponding PSNR values in Table 2 for DwtDctSVD is higher than StegaStamp; whereas in Fig 5., the PSNR of diffusion regeneration on StegaStamp is low. So, I would imagine that the attacked images of StegaStamp will have more visible artifacts but the author did not include any visualization of this. Thus, I consider the current conclusion an overclaim without persuasive argument and suggest the authors provide more evidence to support their claim.

Weakness 3 (Minor) --- inaccurate statements.

In line 1-3 “Invisible watermark safeguards images’ copyrights … prevent people from misusing images … ”.  These are largely believed to be only possible applications that people are thinking of how to use invisible watermarks, but not affirmative conclusions.
Line 4 “ The proposed attack method first adds random noise to an image…”. According to Eq. 1, the proposed methods add noise to the latent feature of the image (attack instance 2 & 3), not the image itself.


[1] An, Bang, et al. "WAVES: Benchmarking the Robustness of Image Watermarks." Forty-first International Conference on Machine Learning.

**Questions:**

1. Similar to Fig. 2, can you also provide (1) watermarks other than DwtDctSVD and (2) curves that are achieved by other regeneration attacks (e.g., VAE, denoising autoencoder)? Current illustration is insufficient to support the claim in the caption “indicating the success of our attack and the validity of the theoretical bound”.

2. As the authors repeatedly claim the superior attack performance of their proposed method, the visualization w.r.t only DwtDctSVD is not sufficient, as DwtDctSVD appears to be the least robust watermark among all the selected watermarks in this paper (see Table 2). Can you provide complete set of attack visualization of all methods on different watermarks considered in this paper? This will also help to clarify "Weakness 2" stated above.

3. As the authors proposed denoising reconstruction (attack instance 1), I think it is necessary to include DiffPure as a baseline attack in the experiment and discuss its performance. This has been applied as a watermark attack method in a published paper [2].

[2] Saberi, Mehrdad, et al. "Robustness of AI-Image Detectors: Fundamental Limits and Practical Attacks." The Twelfth International Conference on Learning Representations. 2023.

**Limitations:**

N/A. The work does not have potential negative impact that needs to be explicitly discussed.

---

> ### Author Rebuttal · Authors · 2024-08-06
>
> Thank you for your thoughtful feedback. We appreciate the opportunity to clarify our contributions and address your concerns.
> > W1: Experiment settings and evaluation are not rigorous. The authors may consider profiling the quality and detectability tradeoff.
>
> We appreciate your suggestion. We have conducted a comprehensive evaluation with various parameter settings for each attacking method. The results are now included in the attached rebuttal PDF.
>
> Specifically, **we have plotted the quality-detectability tradeoff for five watermark schemes across eight different attacking methods**. The x-axis represents quality metrics (SSIM and PSNR, higher is better), while the y-axis shows the detection metric True Positive Rate at a fixed False Positive Rate (TPR@FPR=0.01, lower is better from an attacker's perspective). The strongest attacker should appear in the lower right corner of these plots. We used the same watermark settings as described in the original paper (Section 5, Watermark Setting).
>
> **Key findings**
>
> - Our proposed regeneration attack consistently outperforms other methods across all five watermarking scenarios.
> - For DctDwtSvd, RivaGAN, and SSL watermarks, our regeneration attack instance 2 (VAE) achieves the best Pareto front of quality and attack detectability.
> - Our regeneration attack instance 3 (diffusion model) performs best against Stable Signature and shows strong results across all scenarios. It offers the additional benefit of ease of use and can achieve good attacking results with different noise levels.
>
> These results strongly support our claim that the regeneration attack is a very effective method for attacking various watermarks compared to strong baselines.
>
> > W2: Conclusion with insufficient support. It may be reasonable to argue that StegaStamp can successfully prevent image misuse and does not need a shift as the authors claimed.
>
> We appreciate the reviewer's perspective on StegaStamp. Our conclusion in lines 13-14 is based on both theoretical analysis and empirical evidence:
>
> - Theoretical guarantee: Our regeneration attack is proven to remove certain pixel-based invisible watermarks that perturb the image within a limited range of l2 distance. This guarantee applies to both existing and future watermarking techniques that fall within this category.
> - StegaStamp specifics: As the reviewer noted, StegaStamp introduces "visible artifacts that are enough to raise human caution and suspect they are not original images". Figure 4 shows that the l2 distance for StegaStamp is quite large, making it not "invisible" if we set the l2 distance to be small.
> - Empirical results: To address the reviewer's concerns, we have **included the attacked images for different watermarks, including StegaStamp, in the attached PDF**. These images demonstrate that our regeneration attack (diffusion model) produces high-quality results while successfully evading StegaStamp detection.
>
> Our framework is self-consistent, and we have not hidden any results or overclaimed our conclusions. The additional visual examples provide further support for the effectiveness of our approach across different watermarking techniques.
>
>
> > W3: Inaccurate statements: "Invisible watermark safeguards images' copyrights … prevent people from misusing images … ". These are largely believed to be only possible applications.
>
> We appreciate the reviewer's attention to detail. To clarify, these are not merely possible applications but **are already in use in prominent real-world scenarios**:
>
> - Stable Diffusion [1], a widely used open-source image generation model, employs invisible watermarks to protect generated images.
> - DeepMind [2] utilizes SynthID in images to safeguard copyrights and prevent image misuse.
>
> These examples demonstrate that invisible watermarks are actively being used for copyright protection and preventing image misuse in real applications.
>
>
> > W3: Line 4 "The proposed attack method first adds random noise to an image…". According to Eq. 1, the proposed methods add noise to the latent feature of the image (attack instance 2 & 3), not the image itself.
>
> We thank the reviewer for pointing out this potential source of confusion. To clarify:
>
> The regeneration attack is a family of attacks with multiple instances:
> - Instance 1 adds noise directly to the image.
> - Instances 2 and 3 add noise to the latent feature of the image.
>
> To improve clarity, we will revise the statement to: "adds random noise to an image or its latent feature..."
>
> > Q1
>
> **We have added an example of the RivaGAN watermark with our regeneration attack in the attached PDF** to further illustrate the theoretical guarantee's applicability across different watermarking techniques.
>
> > Q2
>
> We acknowledge the reviewer's point about the need for more comprehensive visualizations. We have now included a complete set of attack visualizations for all methods on the different watermarks considered in our paper. These can be found in the attached PDF.
>
> > Q3
>
> Regarding DiffPure, we appreciate the suggestion to include it as a baseline. While DiffPure is conceptually similar to our regeneration attack instance 3 (diffusion model), we have conducted additional experiments to compare their performance:
>
> Using the RivaGAN watermark as an example, we fixed TPR<0.05 at FPR=0.01 for both regeneration and DiffPure attacks, then compared PSNR:
> - Regen-Diff: PSNR = 23.33
> - DiffPure: PSNR = 23.07
>
> These results demonstrate that our Regeneration-Diffusion approach achieves better image quality while maintaining the same level of watermark removal effectiveness.
>
> ----
> We hope these clarifications and additional results address the reviewer's concerns and further strengthen our paper's contributions. We are grateful for the opportunity to improve our work and would be happy to provide any additional information or clarifications if needed.
>
> ----
>
> [1] https://github.com/Stability-AI/stablediffusion
>
> [2] https://deepmind.google/technologies/synthid/

---

> > ### Comment · Reviewer_V7G3 · 2024-08-07
> > **Reply to the authors**
> >
> > I appreciate the effort that you have put into this rebuttal. Most of my questions and concerns are addressed and I'm inclined to increase my original rating.
> >
> > Nevertheless, I have some further comments and questions regarding the rebuttal provided and hope to be clarified/addressed:
> >
> > 1. DiffPure also seems to have tuning parameters. Thus I believe it is more appropriate to compare with DiffPure by plotting the quality-detectability tradeoff just as what you did to other attacks. I would ask to put this comparison in the same quality-detectability plots.
> >
> > 2. If I remember correctly, you have tried to emphasized two things in your original paper: (1) the proposed regeneration attack is very effective and (2) the regeneration attack powered by diffusion model outperforms the others. However, based on your updated quality-detectability tradeoff, I see that regeneration powered by **VAE** seems to be the best performing one. Is this understanding correct? If so, I would ask the authors to adjust their related comments/conclusions in the paper accordingly.
> >
> > 3. I understand that the authors' emphasis of this paper is the "removability", and the cost of quality loss to remove the watermark is secondary consideration. However, based on the additional results provided in the rebuttal, the cost to remove some watermarks (e.g., StegaStamp) tends to be large (e.g., PSNR $\sim$ 23.33 by Regen-Diff; also the visible flaws shown in the visualization). I hope to see some discussion related on how the "removability" and "cost to remove" trade-off can potentially imply for practical scenarios. For example, how this result support/concerns the application of watermark (copyright protection, preventing image misuse, etc); possible directions to counteract your finding if watermark is not safe let's say; or any idea for the future design of practical watermarks that you can suggest.
> >
> > Overall, if the above points can be further clarified/addressed, I tend to change the rating to accept.

---

> > > ### Author Response · Authors · 2024-08-08
> > >
> > > Thank you for your prompt and insightful feedback! It significantly contributes to the improvement of our paper. We have provided responses and clarifications to your comments below:
> > >
> > > > "DiffPure also seems to have tuning parameters. Thus I believe it is more appropriate to compare with DiffPure by plotting the quality-detectability tradeoff just as what you did to other attacks. I would ask to put this comparison in the same quality-detectability plots."
> > >
> > > We appreciate your suggestion. We have now obtained all the results for DiffPure. In summary, DiffPure performs comparably to Regen-Diff, albeit with a slightly worse quality-detectability tradeoff. Due to the inability to edit the rebuttal PDF, we will include this comparison in the revised paper. Additionally, we offer the following analysis:
> > >
> > > - Mathematically, DiffPure and Regen-Diff employ the same method, adding noise to samples via the forward process with a small diffusion timestep, and then solving the reverse VP-SDE to recover clean samples.
> > >
> > > - DiffPure, or [1]'s implementation, utilizes the 256x256 diffusion (unconditional) checkpoint from the guided-diffusion library [2] pretrained on ImageNet data. In contrast, we use the stable-diffusion-2-1 latent diffusion model from Stable Diffusion, pretrained on the LAION-5B dataset, which offers superior generation quality. Our implementation, as demonstrated in the Supplementary Material, supports many other latent diffusion models.
> > >
> > > > "If I remember correctly, you have tried to emphasized two things in your original paper: (1) the proposed regeneration attack is very effective and (2) the regeneration attack powered by diffusion model outperforms the others. However, based on your updated quality-detectability tradeoff, I see that regeneration powered by VAE seems to be the best performing one. Is this understanding correct? If so, I would ask the authors to adjust their related comments/conclusions in the paper accordingly."
> > >
> > > Thank you for your feedback! For DctDwtSvd, RivaGAN, and SSL watermarks, Regen-VAE achieves the best Pareto front of quality and attack detectability. Regen-Diff performs best against Stable Signature and shows strong results across all scenarios. It also offers ease of use and achieves good attacking results with different noise levels. We will adjust our related comments and conclusions in the revised paper accordingly.
> > >
> > > > "I hope to see some discussion related on how the "removability" and "cost to remove" trade-off can potentially imply for practical scenarios. For example, how this result support/concerns the application of watermark (copyright protection, preventing image misuse, etc); possible directions to counteract your finding if watermark is not safe let's say; or any idea for the future design of practical watermarks that you can suggest."
> > >
> > > We appreciate your suggestions and feedback! In the revised paper, we will discuss how the "removability" and "cost to remove" trade-off can potentially apply to practical scenarios. For example, from the watermark addition side, if they can increase the L2 distance without significantly altering image perception quality, it could be practical. Researchers can use regeneration as an attacking baseline; if all attacked images have lower quality than a set threshold, the watermark can be deemed sufficient. Overall, we advocate for using semantic watermarks. As stated in response to reviewer n9vc, we are exploring methods to enhance the robustness of post-hoc watermarking. One promising approach involves using powerful image editing techniques to add or remove unimportant subjects or alter textures. These modifications are visible but appear as normal image content without the watermark key.
> > >
> > > ----
> > > Thank you again for your feedback and prompt response. We hope these additional results address your concerns.
> > >
> > >
> > > ----
> > > [1] Saberi et al. "Robustness of AI-Image Detectors: Fundamental Limits and Practical Attacks." ICLR 2024.
> > >
> > > [2] https://github.com/openai/guided-diffusion

---

> > > > ### Comment · Reviewer_V7G3 · 2024-08-08
> > > > **Reply to the authors**
> > > >
> > > > Thank you for the prompt response.
> > > >
> > > > All of my concerns are sufficiently addressed. Therefore, I have decided to change my rating to accept.

---

> > > > > ### Author Response · Authors · 2024-08-08
> > > > >
> > > > > Thank you for your active engagement and insightful contributions!
> > > > > We will make sure to revise our manuscript according to our discussion.

---

### Official Review · Reviewer_n9vc · 2024-07-13

**Soundness:** 3
**Presentation:** 3
**Contribution:** 3
**Rating:** 7
**Confidence:** 3

**Summary:**

This paper proposes regeneration attacks, which adds destructive Gaussian noise to the latent representation of the watermarked image, and then reconstructs the corrupted latent to reconstruct the original clean image. The paper provides theoretical guarantee that shows the trade-off function between the Type I error and the Type II error after the attack, in addition to the theorem that shows the ability of the regeneration attack's capability to produce images with similar quality as the generative model that reconstructs the corrupted latent. The paper also introduces a potential defense mechanism that can survive the regeneration attack.

**Strengths:**

The paper is well-written and easy to read. The presentation is clear and the empirical result is thorough and informative. Besides empirical results, the author also provides theoretical guarantees on the claim. Besides proposing the regeneration attack, the author also provides a potential defense mechanism that sheds light on future watermarking research under the proposed attack.

**Weaknesses:**

The proposed method relies on the upper bound $L \geq L_{x,w}$ to effectively calibrate $\sigma$, which is slightly unrealistic considering the embedding function $\phi$ and original image $x$ is unknown to the attacker, even though a uniform upper bound may exist, this could potentially affect the attack performance or image quality since the attacker normally has no access to the decoding scheme thus unable to verify the performance of the attack while maintaining the image quality.

**Questions:**

Major concerns are already addressed in Appendix A, so here are some minor questions that may be slightly out of the scope.
1. For in-processing watermarking, semantic watermarking is a great alternative for preserving visual quality as well as robustness to regeneration attacks. However, for post-hoc watermarking, changing the semantic content leads to unsatisfactory results. How do you envision balancing the robustness of semantic watermarks with their increased visibility in practical applications? Are there any strategies you are exploring to minimize the visual impact while maintaining robustness under regeneration attacks?
2. The Stegastamp method seems to be able to withstand the proposed attack due to a relatively high l2 distance in both pixel and latent space. Figure 5 shows that with greater noise levels, StegaStamp fails to withstand the regeneration attacks, but the reduced PSNR and SSIM indicate significant degradation in image quality. Since StegaStamp's image quality is already low, enforcing regeneration attacks that further degrade image quality seems unrealistic in real-world scenarios. Given that PSNR and SSIM are pixel-level metrics, such degradation is expected after multiple perturbations in the semantic space. I wonder if the semantically meaningful content is still preserved after regeneration attacks capable of breaking StegaStamp. Specifically, a visualization for Figure 5 or an LPIPS curve might be helpful for understanding the preservation of semantic content.

-------------------Post rebuttal Edit------------------------

I appreciate the author for the comments. My questions has been well addressed. I have increased my rating accordingly.

**Limitations:**

potential negative societal impact has not been addressed but the author does provide potential defense mechanisms for the proposed attack method.

---

> ### Author Rebuttal · Authors · 2024-08-06
>
> Thank you for your valuable feedback. We appreciate the opportunity to address the specific points raised and provide further clarification.
>
> ### 1. Calibration of $\sigma$ and Attack Performance
>
>    > "Relies on the upper bound $L \geq L_{x,w}$ to effectively calibrate $\sigma$… this could potentially affect the attack performance or image quality since the attacker normally has no access to the decoding scheme, thus unable to verify the performance of the attack."
>
>    Access to the decoding scheme is not necessary for calibrating $\sigma$. Although we cannot calibrate by verifying if an attack is successful, we can still choose the optimal $\sigma$ through a binary search. Typically, the quality of the attacked image degrades as $\sigma$ increases, allowing us to use binary search to find the largest $\sigma$ that maintains acceptable image quality. Additionally, we discuss controlling a certain degree of Certified Watermark Freeness (CWF) a priori in Appendix lines 522-529. Here is the original content for your reference:
>
> - "Our guarantee in Theorem 1 depends on the specific watermark injected into a specific image instance through the unknown local Lipschitz parameter $L_{x,w}$. Specifying a fixed CWF level requires a uniform upper bound of $L_{x,w}$ independent of $x$ and $w$. For example, when the embedding $\phi$ is trivial (identity map), we can take $L_{x,w} \leq 1$. When $\phi$ involves lower pass filtering, such as a Fourier transform that removes all high-frequency components except the top \(k\) dimensions, we can bound $L_{x,w} \leq \sqrt{k}/n$, where $n$ is the number of pixels. Generally, any linear transformation with a bounded operator norm is suitable."
>
> ### 2. Balancing Robustness and Visibility of Semantic Watermarks
>
>    > "How do you envision balancing the robustness of semantic watermarks with their increased visibility in practical applications? Are there any strategies you are exploring to minimize the visual impact while maintaining robustness under regeneration attacks?"
>
>    We appreciate your interest in future developments. Yes, we are actively exploring methods to enhance the robustness of post-hoc watermarking. One promising approach involves leveraging powerful image editing techniques to add or remove unimportant subjects or alter textures. These modifications are visible but appear as normal image content without the watermark key.
>
> ### 3. Visualization or LPIPS Curve for Figure 5
>
>    > "Specifically, a visualization for Figure 5 or an LPIPS curve might be helpful for understanding the preservation of semantic content."
>
> We have included a visualization image in the Rebuttal PDF. This visualization demonstrates that the semantically meaningful content remains preserved.
>
> ----
> Overall, we hope these clarifications address your concerns and demonstrate the strength and potential of our approach. We appreciate your consideration and look forward to your stronger support.

---

### Author Rebuttal · Authors · 2024-08-07

To address reviewer n9vc's question 2 and reviewer V7G3's several concerns, we have included more figures in the attached PDF.


## Figure 1: Quality-Detectability Tradeoff


During the rebuttal, we conducted a comprehensive evaluation with various parameter settings for each attacking method.


For Figure 1, we plotted the quality-detectability tradeoff for five watermark schemes across eight different attacking methods. The x-axis represents quality metrics (SSIM and PSNR, higher is better), while the y-axis shows the detection metric True Positive Rate at a fixed False Positive Rate (TPR@FPR=0.01, lower is better from an attacker's perspective). The strongest attacker should appear in the lower right corner of these plots. We used the same watermark settings as described in the original paper (Section 5, Watermark Setting)


**Attack Methods and Parameters**

*Regeneration Attacks:*
- Diffusion model: noise steps {10, 30, 50, 100, 150, 200}
- VAE-Cheng2020: compression factors {1, 2, 3, 4, 5, 6}
- VAE-Bmshj2018: compression factors {1, 2, 3, 4, 5, 6}

*Baseline Attacks:*
- JPEG compression: quality {10, 20, 30, 40, 50, 60}
- Gaussian blur: kernel size {2, 4, 6, 8, 10, 12}
- Brightness enhancement: change of {2, 4, 6, 8, 10, 12}
- Gaussian noise: standard deviation {5, 10, 15, 20, 25, 30}
- Contrast enhancement: change of {0.5, 2, 3, 4, 5, 7}

**Key findings**
- Our proposed regeneration attack consistently outperforms other methods across all five watermarking scenarios.
- For DctDwtSvd, RivaGAN, and SSL watermarks, our regeneration attack instance 2 (VAE) achieves the best Pareto front of quality and attack detectability.
- Our regeneration attack instance 3 (diffusion model) performs best against Stable Signature and shows strong results across all scenarios. It offers the additional benefit of ease of use and can achieve good attacking results with different noise levels.

These results strongly support our claim that the regeneration attack is a very effective method for attacking various watermarks compared to strong baselines.




## Figure 2-4: Additional Visualizations
- Figure 2: More visualizations of attacked images on RivaGAN watermark. All regeneration attacked images can evade detection.
- Figure 3: More visualizations of attacked images on SSL watermark. All regeneration attacked images can evade detection.
- Figure 4: More visualizations of attacked images on StegaStamp watermark. The diffusion model of the regeneration attacked images can evade detection.

## Figure 5: Theoretical and Empirical Trade-off Functions


Figure 5 shows another example similar to Figure 2 in our original paper. It presents theoretical and empirical trade-off functions for RivaGAN watermark detectors after our attack. Trade-off functions indicate how much less Type II error (false negative rate) the detector gets in return by having more Type I error (false positive rate). Theoretically, after the attack, no detection algorithm can fall in the *Impossibility Region* and have both Type I error and Type II error at a low level. Empirically, the watermark detector performs even worse than the theory, indicating the success of our attack and the validity of the theoretical bound. We use 500 watermarked MS-COCO images with an empirically valid upper bound of $L=1$ and noise level $\sigma = 0.57\Delta$.


----
We believe these additional evaluations and visualizations further enhance our paper's quality and address the reviewers' concerns.

---

### Decision · Program_Chairs · 2024-09-25

**Decision:**

Accept (poster)

**Comment:**

After discussion, all the reviewers provided positive feedback on this submission, therefore we recommend accepting it.